# Activation energy and force fields during topological transitions of fluid lipid vesicles

Matteo Bottacchiari [1], Mirko Gallo[1,2], Marco Bussoletti [1] & Carlo Massimo Casciola [1✉]

Topological transitions of fluid lipid membranes are fundamental processes for cell life. For example, they are required for endo- and exocytosis or to enable neurotransmitters to cross the neural synapses. Here, inspired by the idea that fusion and fission proteins could have evolved in Nature in order to carry out a minimal work expenditure, we evaluate the minimal free energy pathway for the transition between two spherical large unilamellar vesicles and a dumbbell-shaped one. To address the problem, we propose and successfully use a Ginzburg-Landau type of free energy, which allows us to uniquely describe without interruption the whole, full-scale topological change. We also compute the force fields needed to overcome the involved energy barriers. The obtained forces are in excellent agreement, in terms of intensity, scale, and spatial localization with experimental data on typical fission protein systems, whereas they suggest the presence of additional features in fusion proteins.

[1] Department of Mechanical and Aerospace Engineering, Sapienza Università di Roma, Rome, Italy. [2] Present address: School of Architecture, Technology and Engineering, University of Brighton, Brighton, United Kingdom. ✉email: carlomassimo.casciola@uniroma1.it

Topological transitions of fluid lipid membranes are involved in most of the fundamental processes of cell life, like endocytosis and exocytosis. An example of such transformation is the merging of two membranes. This is the case of vesicle–vesicle fusion or viral membrane fusion. Indeed, viruses enveloped by a lipid bilayer, such as HIV, Ebola virus, influenza, measles, rabies virus, and SARS-CoV-2 can infect a target cell by fusion of their membrane with the cell plasma membrane[1,2]. Viral infection can also occur via endocytosis, in which the plasma membrane undergoes fission to internalize the virus via an endosome. Therefore, another important topological change is membrane fission, which is also fundamental for cell division and therefore for life[3,4]. Topological transitions of lipid membranes are of great interest not only in biology and biophysics but also in medicine and in the pharmaceutical industry. Indeed, lipid-based nanoparticles are used for drug delivery, offering many advantages including biocompatibility, bioavailability, self-assembly, and payload flexibility[5]. Micelles, closed lipid monolayers, are currently used in mRNA vaccines against COVID-19 and many other lipid nanoparticle-mRNA applications are under clinical evaluation, e.g. for the treatment of cancer or genetic diseases[6]. Regardless of the application, all these nanoparticles are engineered to overcome the physiological barriers by exploiting topological transitions[7].

Fluid lipid membranes can be mechanically described using a continuum approach initially introduced by Canham[8] and Helfrich[9]. Such a classical elastic perspective describes a membrane as a two-dimensional surface $\Gamma$ with an energy density depending on its principal curvatures. An expansion of this density up to the second-order in curvatures leads to the Canham–Helfrich Hamiltonian:

$$E_{\mathrm{CH}}[\Gamma] = 2k \int_{\Gamma} (M - m)^2 \, \mathrm{d}S + k_{\mathrm{G}} \int_{\Gamma} G \, \mathrm{d}S. \tag{1}$$

Here, the first term on the right-hand side is the bending energy and the second one is the Gaussian energy. $M$ is the mean curvature of the surface, $G$ it's Gaussian curvature, $m$ a spontaneous mean curvature that the membrane tends to adopt in absence of external forces, and $k$ and $k_{\mathrm{G}}$ are called bending rigidity and Gaussian curvature modulus, respectively. $k$ can be experimentally measured in different ways[10], whereas $k_{\mathrm{G}}$ is more elusive due to the celebrated Gauss–Bonnet theorem

$$\int_{\Gamma} G \, \mathrm{d}S = 2\pi\chi(\Gamma) - \int_{\partial\Gamma} k_{\mathrm{g}} \, \mathrm{d}l, \tag{2}$$

where $\chi(\Gamma)$ is the Euler characteristic of $\Gamma$ and $k_{\mathrm{g}}$ is the geodesic curvature of the surface boundary $\partial\Gamma$. In the vesicle case, since they are compact surfaces without boundary, the line integral vanishes, and $\chi(\Gamma)$ becomes equal to $2-2g$, being $g$ the genus of the surface. Therefore, the Gaussian energy term remains constant as long as no topological transitions occur, leading to the aforementioned elusive behavior of $k_{\mathrm{G}}$. A stability argument[11] shows that $-2 < k_{\mathrm{G}}/k < 0$ and there is evidence[12–14] that $k_{\mathrm{G}} \approx -k$. Because of the scale invariance of the Canham–Helfrich free energy, for a given topology, vesicles shapes are dictated by their reduced volume $v = V/(\pi D_{\mathrm{ve}}^3/6)$, as well as by their reduced spontaneous curvature $m_0 = m D_{\mathrm{ve}}$, where $D_{\mathrm{ve}} = \sqrt{A/\pi}$ is the characteristic length of the vesicle under consideration, having area $A$ and volume $V$. The Canham–Helfrich description is thought to hold[15] for vesicles with a characteristic length $D_{\mathrm{ve}} \geq 40\ell_{\mathrm{me}}$, being $\ell_{\mathrm{me}}$ the lipid bilayer thickness, which is usually about 5 nm; otherwise, higher-order terms in the energy density could make a significant contribution. In fact, for symmetric membranes[15–17], this limit safely reduces to $10\ell_{\mathrm{me}}$. As a matter of fact, most of the experimental results concerning lipid bilayers are still today interpreted in light of this celebrated model.

Nevertheless, its main limitation consists of the inability to account for topological changes, like those associated with fusion and fission processes.

The most commonly used techniques for in silico studies of topological transitions to date are coarse-grained molecular dynamics (MD) and dissipative particle dynamics (DPD)[18–28]. These computer simulations, which take into account the molecular details of lipid bilayers, allow monitoring in time morphological changes of small liposomes[29]. In many cases of interest, including topological transitions, the size of the vesicles is significantly larger or the characteristic time of the process is longer than accessible to purely atomistic methods. For example, in neurotransmission, the fusion of small synaptic vesicles can take hundreds of microseconds[30]. Recently, in order to reach larger vesicle sizes, hybrid, multiscale approaches have been proposed[31–33]. Describing the complete topological rearrangement of these large-sized vesicles is the target of the present study.

Concerning experiments, controlled fission of cell-sized vesicles by low densities of membrane-bound proteins has been recently reported[34] and other examples of fission experiments can be found in literature[35,36]. As regards fusion, the merging of giant liposomes has been observed[37–39] along with the stalk intermediate[40] and activation energies for small liposomes fusion events have been measured by means of kinetic analysis[41,42].

In order to enable the modeling of full-scale topological transition, a Ginzburg–Landau approach[43], as opposed to the classical Canham–Helfrich description, is intrinsically able to handle topological transformations. In this context, an analog of the bending energy term was initially introduced by Du et al.[44–46], leading to numerous applications regarding, e.g., vesicle adhesion, equilibrium shapes, pearling instability, or red blood cells under flow[47–55]. Furthermore, it has been pointed out[56] that it is possible to retrieve topological information from such models. However, all these works do not include the Gaussian contribution to free energy in the dynamics. As will be shown below, the inclusion of a term accounting for such a contribution is crucial to correctly predict the physics of fusion and fission events. Indeed, in accordance with the Gauss–Bonnet theorem, the term we introduce allows for the quantized energy jumps that significantly contribute to the free energy barriers of topological transitions. From a strictly mathematical point of view, the proposed free energy function regularizes the Gaussian term of the Canham–Helfrich Hamiltonian, allowing the description of the process also across the topological change.

In this work, we develop, numerically demonstrate, and use a Ginzburg–Landau type of free energy to study fission and fusion events of large unilamellar vesicles (LUVs). Exploiting rare event techniques[57], we compute a minimal energy pathway (MEP)[58–61] and the free energy barrier between two spherical vesicles and a dumbbell-shaped one, a case recently observed in experiments[34]. We also compute the force fields needed to overcome these barriers in a straightforward manner, uniquely accounting for the force component arising from the Gaussian energy. These forces are necessary to balance the reaction resulting from the membrane (bending and Gaussian) elasticity and incompressibility; see Guckenberger et al.[62] for a discussion on the difficulties in computing the bending forces using more classical approaches. The computation of the complete system of forces is expected to pave the way for exploring how the protein machinery effectively works across the full scale of vesicles.

In the main text, for the reason of brevity and definiteness, we focus on the aforementioned system considering a zero spontaneous curvature. However, we stress that the approach we propose and the simulations that can be carried out thereof are completely general, as demonstrated by some further examples illustrated in the Supplementary Discussion.

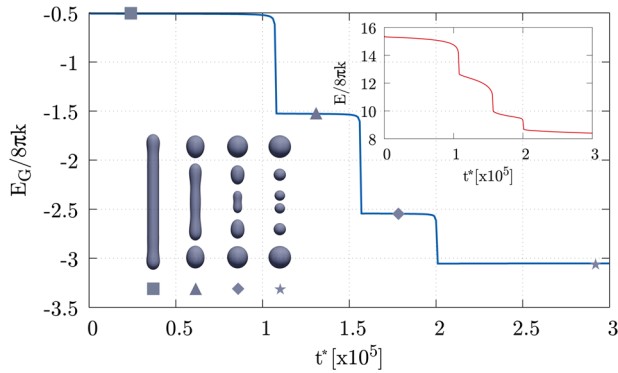

**Fig. 1 Free energy evolution example.** The phase-field Gaussian energy $E_G$ during a series of scissions of a prolate shape into several spheres (main plot, blue line), where symbols identify the shown vesicle configurations. The energy jumps by $-4\pi k$ for any division as prescribed by the Gauss–Bonnet theorem ($k = -k_G$). The fission process occurs due to the presence of a spontaneous curvature $m^\star \approx 0.42$. Time evolution is given by the Allen–Cahn gradient flow with $M^\star = 8$ (see the section "Methods" for more details on the dynamics, the adopted numerical scheme, and dimensionless quantities). The inset shows the total energy $E = E_B + E_G$, which monotonically decreases in time, revealing the stability of the scheme. This $z$-axial symmetric simulation has been carried out in a $[0, 36] \times [0, 440]$ computational domain in the $r^\star$–$z^\star$ plane with a $54 \times 660$ mesh, initial $D_{ve}^\star = 1/\lambda \approx 109$ and $dt^\star = 4$. There is no constraint on the area, which, at the end of the simulation, differs from the initial value by ~6.9%. Volume is conserved with a relative error smaller than $10^{-7}$ with respect to its initial value.

## Results and discussion

**Free energy functional.** The classical Canham–Helfrich model succeeds in describing many aspects of the vesicle dynamics but rules out the possibility of dealing with topological changes unless surgical operations are conceived to cut and paste patches of the membrane[63]. A viable alternative to the sharp interface description is to employ a smooth function defined on a domain $\Omega$ —the phase-field $\phi(\mathbf{x})$— that discriminates between the inner and the outer environment of the vesicle assuming the limiting values $\pm 1$ in the two regions. The $\phi(\mathbf{x}) = 0$ level set represents the membrane mid-surface $\Gamma$. The transition between the two limiting values takes place in a narrow region whose width is controlled by a small parameter $\epsilon$. This region will also be related to the thickness of the lipid bilayer. The main advantage of describing the membrane with such a field lies in the fact that it enables topological modifications of the membrane, allowing to address the problem of vesicle fusion and fission.

A free energy functional

$$E[\phi] = E_B[\phi] + E_G[\phi], \tag{3}$$

is associated with each field configuration, where

$$E_B[\phi] = k \frac{3}{4\sqrt{2}} \epsilon \int_\Omega \Psi_B^2 \, dV, \tag{4}$$

$$\Psi_B = \nabla^2\phi - \frac{1}{\epsilon^2}(\phi^2 - 1)(\phi + \sqrt{2}\epsilon m), \tag{5}$$

and

$$E_G[\phi] = k_G \frac{35}{16\sqrt{2}} \epsilon^3 \int_\Omega \Psi_G \, dV, \tag{6}$$

$$\Psi_G = \frac{\nabla|\nabla\phi|^2 \cdot \nabla|\nabla\phi|^2}{2} - (\nabla|\nabla\phi|^2 \cdot \nabla\phi)\nabla^2\phi$$
$$+ |\nabla\phi|^2 \left[ (\nabla^2\phi)^2 + \nabla\phi \cdot \nabla\nabla^2\phi - \frac{\nabla^2|\nabla\phi|^2}{2} \right]. \tag{7}$$

$E_B[\phi]$ models the bending energy of the membrane[44], while $E_G[\phi]$ is the term proposed here to account for the Gaussian contribution. In the section "Methods", we show that the free energy functional $E[\phi]$ recovers the Canham–Helfrich Hamiltonian, $E[\phi] \sim E_{CH}[\Gamma]$, in the sharp interface limit ($\lambda = \epsilon/D_{ve} \ll 1$).

Furthermore, lipid vesicles are subjected to geometrical constraints on area $A$ and enclosed volume $V$. Indeed, given the large tension associated with the area change, membrane bending cannot significantly modify $A$. The volume $V$ is instead determined by the osmotic conditions. In order to enforce the above constraints in this phase-field context, we use suitable functionals $A[\phi]$ and $V[\phi]$ which recover the vesicle area and volume, respectively, in the sharp interface limit (see the section "Methods").

Throughout the paper, an asterisk will denote the dimensionless quantities obtained using $\epsilon$ as the reference length and $8\pi k$ as the reference energy. The latter is the bending energy of an isolated sphere. The typical value of the bending rigidity is $k = 20\, k_B T$, with $k_B$ the Boltzmann constant and $T$ the temperature. Moreover, unless, otherwise explicitly stated, we will henceforth assume $k_G = -k$.

As an illustrative example of the effectiveness of the approach, Fig. 1 shows the Gaussian energy during a series of scissions of an unstable prolate shape into several spheres due to the presence of a spontaneous curvature, see also Rueda-Contreras et al.[64]. The evolution equation is described in the section "Methods" together with the adopted numerical scheme. In the same section, the consistency of the present phase-field approach with the Gauss–Bonnet theorem is discussed. Here, it is only worth saying that the energy functional we propose is able to properly capture the Gaussian energy jumps due to topological transitions.

**Minimal energy pathway.** In the topological transition between two spherical vesicles and a dumbbell-shaped one, which are two stable states, the system goes through a sequence of configurations $\phi_\alpha(\mathbf{x})$ in the space of the phase field, identifying a path which we parameterize by the normalized arc-length $\alpha \in [0, 1]$. An MEP for this transition is a curve on the energy landscape $E[\phi]$ connecting the two stable states $\phi_{\alpha=0}(\mathbf{x})$ and $\phi_{\alpha=1}(\mathbf{x})$, respectively, and such that it is everywhere tangent to the gradient of the potential ($\partial\phi_\alpha/\partial\alpha \propto \delta E[\phi_\alpha]/\delta\phi$), except at critical points[65]. An initial guess of the path is discretized in a *string* made up of $N = 100$ images corresponding to $\alpha_i = (i-1)/(N-1)$. The initial guess is relaxed towards the MEP by means of the string method[57,66,67], suitably accounting for the constraints of constant total surface area, Eq. (17), and enclosed volume, Eq. (18) (see the section "Methods"). The obtained MEP goes through a saddle point $\phi_{\alpha_c}(\mathbf{x})$ for the free energy, determining the transition barriers $\Delta E_{0\to1}^\dagger = E[\phi_{\alpha_c}] - E[\phi_{\alpha=0}]$ and $\Delta E_{1\to0}^\dagger = E[\phi_{\alpha_c}] - E[\phi_{\alpha=1}]$, for the forward and backward processes, respectively.

Figure 2 shows the computed MEP for membranes with zero spontaneous curvature, $m = 0$. Since the phase-field $\phi$ reaches its limiting values $\pm 1$ with an accuracy of about 3% already at a distance of $\pm 3\epsilon$ from the $\phi = 0$ membrane mid-surface, we assume that $\ell_{pf} = 6\epsilon$ represents the thickness of the diffuse interface. In the section "Methods", we show that the phase-field description recovers the Canham–Helfrich model in the limit of small $\lambda \propto \ell_{pf}/D_{ve}$. Our numerical experiments, reported in the Supplementary

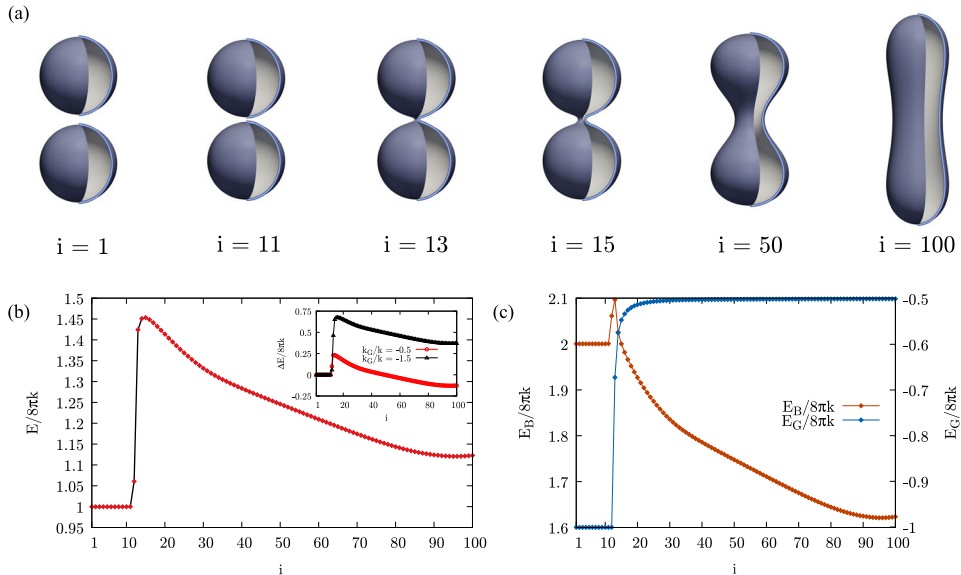

**Fig. 2 The minimal free energy path.** The MEP connecting two spheres of radius $R^\star = 87.5$ with a dumbbell shape, $k = -k_G$. The path consists of vesicles with a constant area and volume and therefore with constant reduced volume $v \approx 1/\sqrt{2}$. There is no spontaneous curvature, $m^\star = 0$. This $z$-axial symmetric result is obtained with the string method and the here proposed free energy functional, using a $[0, 96] \times [-245, 245]$ computational domain in the $r^\star - z^\star$ plane with a grid of $144 \times 735$ nodes per image, $N = 100$ images and $1/\lambda \approx 247.5$. As explained in the text, this setting leads to having $D_{ve} \approx 206$ nm. The dimensionless quantities are however useful because, far from the moment in which the topology changes, the scale invariance is expected to hold. **a** Six vesicle shapes along the minimal energy path, identified by their image number $i = (N-1)\alpha_i + 1$, being $\alpha$ the string parameter (equal arc-length parameterization). From right to left we can observe the fission process of the dumbbell shape into two spheres, whereas from left to right the fusion process. **b** The free energy (Eq. (3)), along the path. The saddle point is placed between the images $i = 14$ and $i = 15$ and consists of two spherical vesicles connected by a catenoid-like neck. The black solid line identifies the steepest stretch of the MEP, namely the region where a refinement will be carried out in the Discussion. The inset shows two additional MEPs for the same system but with $k_G/k = -0.5$ and $k_G/k = -1.5$, respectively. In this case, the energy difference with respect to the initial state, $\Delta E$, is reported on the y-axis. **c** The bending and Gaussian energy contributions along the path.

Methods, point out that this asymptotic behavior is already achieved when $\ell_{pf}/D_{ve} \gg (\ell_{me}/D_{ve})_{max} = 1/40$.

Since the relative distance between approaching membrane segments is relevant during the topological transition, it is crucial that the diffuse interface width matches the bilayer thickness. This requirement fixes the scale of our system. Setting $\ell_{pf} = \ell_{me} = 5$ nm, the configurations shown in Fig. 2a correspond to vesicles with $D_{ve} \approx 206$ nm, thus within the range of validity of the asymptotic Canham–Helfrich model.

Figure 2a shows successive configurations along the MEP. Increasing/decreasing $\alpha$ corresponds to moving along the path in the direction of the fusion/fission (forward/backward) process, respectively. Proceeding forward, the two vesicles come closer to each other without deforming, get in touch, and merge together forming a narrow neck that expands until the final dumbbell-shaped configuration is reached. As explained in the section "Introduction", the equilibrium states of a vesicle are determined by its reduced volume and reduced spontaneous curvature, which, in the present case, are $v = 1/\sqrt{2}$ and $m_0 = 0$, respectively, where $1/\sqrt{2}$ is the only reduced volume compatible with a vesicle obtained from the fusion of two spheres of the same radius (see the Supplementary Discussion). With these parameters, it is possible to reach two axisymmetric configurations with the topology of a sphere, namely one oblate-discocyte shape and one prolate-dumbbell shape[68]. The latter has the lowest energy and, in the present case, is the equilibrium state assigned to the string as the final configuration, $\phi_{\alpha=1}(\mathbf{x})$.

The main plot of Fig. 2b shows the free energy profile along the MEP. The free energy of the final configuration (prolate) is $E[\phi_{\alpha=1}]/(8\pi k) \approx 1.12$, which is larger than the sum $E[\phi_{\alpha=0}]/(8\pi k) = 1$ of the initial energies of the two spheres. Both values are in excellent agreement with the data reported in the

literature[68]. One may notice that the two-sphere configuration possesses a sequence of neutral equilibrium states, corresponding to rigid translations during which the two vesicles approach/separate from each other (configurations $i$ from 1 to 11, as also depicted in Fig. 2a). The saddle point consists of two spheres connected by a small narrow neck and is located between configurations $i = 14$ and $i = 15$, with the latter having the highest energy of the two, $E[\phi_{\alpha=\alpha_c}]/(8\pi k) \approx 1.45$. It should be noticed that such a configuration possesses the bending energy of two spheres together with the Gaussian energy and the topology of a single sphere. Hence, the forward and backward free energy barriers are $\Delta E^\dagger_{0 \to 1}/(8\pi k) \approx 0.45$ and $\Delta E^\dagger_{1 \to 0}/(8\pi k) \approx 0.33$, respectively. Considering a bending rigidity[10] $k$ of order 20 $k_B T$, it turns out that, in the present conditions, $m = 0$, both fusion and fission processes cannot take place spontaneously and require further agents in order to happen, in addition to the elasticity and thermal fluctuations. These agents are typically protein systems whose mode of operation may differ considerably from case to case, e.g. by involving active motors or simply modifying the membrane spontaneous curvature[69]; see Fig. 1 and the Supplementary Discussion for examples of spontaneous curvature-induced topological rearrangement. Still, in the main plot of Fig. 2b, it is possible to observe a substantial asymmetry between fusion and fission, with a much steeper energy increase required to reach the transition state in the fusion process.

Figure 2c provides the bending and Gaussian contributions to the free energy along the MEP. Apparently, the forward barrier $\Delta E^\dagger_{0 \to 1}$ is almost entirely due to the Gaussian energy jump associated with the topological change. On the other hand, the backward barrier $\Delta E^\dagger_{1 \to 0}$ builds up continuously with the progressive deformation of the prolate shape to form the narrow neck preceding the actual fission. This conclusion is substantiated

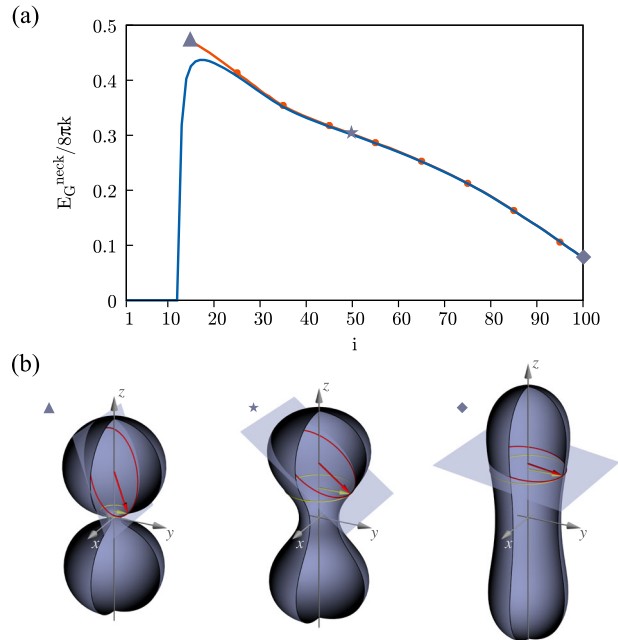

**Fig. 3 Neck Gaussian energy. a** Normalized Gaussian energy of the neck, $E_G^{neck}(Z)/(8\pi k)$ (Eq. (8)), along the MEP (blue line). The orange line with dots provides the neck Gaussian energy as post-processing based on the sharp interface Canham–Helfrich energy (Eq. (1)), computed considering the $\phi = 0$ levels set as the membrane mid-surface. The agreement between the two curves progressively deteriorates when getting closer to the saddle point, due to the increasing curvature of the membrane generatrix. In this region, the finite thickness of the bilayer plays a crucial role and is taken into account by the phase field. **b** Three membrane configurations sketching the upper (yellow) circular boundary of the neck with its curvature radius $r(Z)$ (yellow arrow), and the osculating (red) circle to the vesicle cross-section with the cutting plane passing through the neck boundary and containing both the surface normal and the tangent to the circle. The radius $R_n$ of the osculating circle is shown as a red arrow. Using these quantities, the sharp interface Canham–Helfrich energy reads $E_{G,CH}^{neck}(Z)/(8\pi k) = \sqrt{1 - (r(Z)/R_n)^2}/2$. The position of each configuration along the MEP is denoted by the corresponding symbol (triangle, star, and rhombus).

in the inset of Fig. 2b, where additional MEPs with different $k_G$ show that the fusion barrier is directly affected by the Gaussian modulus while the fission one remains substantially unaffected.

The formation of the catenoid-like neck[70] has also been observed in the experiments[36]. Operationally, we define the neck region as the $z$-chunk of the fused vesicle where the local contribution to the Gaussian energy

$$E_G^{neck}(Z) = k_G \frac{35}{16\sqrt{2}} \epsilon^3 \int_{-Z}^{+Z} dz \int 2\pi r\, \psi_G\, dr, \qquad (8)$$

is positive. The Gaussian energy of the neck along the MEP is shown in Fig. 3a, blue line. Proceeding from left to right, $E_G^{neck}(Z)/(8\pi k)$ sharply increases to a value close to (though smaller than) 0.5 and subsequently decreases. According to the Canham–Helfrich model, the sharp interface Gaussian energy of a sphere is $E_G^{CH}/(8\pi k) = -0.5$. Given two initially disjoint sharp spheres ($E_G^{CH}/(8\pi k) = -1$), a joining neck changes the topology and reduces the energy to that of a single sphere, $E_G^{CH}/(8\pi k) = -0.5$. There are two main reasons why the present free energy provides a neck contribution that is slightly smaller than 0.5: (i) close to the transition state, the curvature of the neck generatrix is comparable with the finite thickness of the bilayer so that the sharp-interface model is inappropriate; (ii) the value 0.5

is an upper limit for the sharp interface Gaussian energy of the neck (see, e.g., the Gauss Map[71]). Evidently, $E_G^{neck}(Z)$ is the main contribution to the forward barrier $\Delta E_{0\to1}^\dagger/(8\pi k) \approx 0.45$. Proceeding to the right along the MEP, beyond the saddle point, $E_G^{neck}(Z)$ progressively decreases, Fig. 3a, blue line. Since, Fig. 2c, in that region the total Gaussian energy remains overall constant, $E_G/(8\pi k) = -0.5$, the (Gaussian) energy lost by the neck is redistributed to the remaining, dome-like parts of the vesicle. Figure 3a, orange line with dots, also provides the neck Gaussian energy as post-processing based on the sharp interface Canham–Helfrich energy (Eq. (1)), computed considering the $\phi = 0$ levels set as the membrane mid-surface (see Fig. 3b and the Supplementary Methods for additional details).

**Force fields**. Figure 4 focuses on the region of the MEP where the most relevant events associated with the topological transition take place, images $i = 11, \ldots, 40$. The contour plots provide the structure of the phase-field as a function of radius $r^*$ and axial coordinate $z^*$, with $\phi$ smoothly joining the inner region $\phi = 1$ to the outer region $\phi = -1$ through the layer of dimensionless thickness $\ell_{pf}^* = 6$.

As explained in the section "Methods", each image of the string can be rendered a state of equilibrium by introducing a force field $\mathbf{f} = -\delta E/\delta\phi\nabla\phi$ that counterbalances the membrane elastic reaction. Considering the forward transition, $0 \to 1$, such force field from $\alpha = 0$ to $\alpha = \alpha_c$ can be interpreted as the external force needed to drive the transition under quasi-static conditions, thus spending the minimal work $\mathcal{W}_{0\to1} = \Delta E_{0\to1}^\dagger$. Once the critical state is overcome, the system can be left to evolve spontaneously until it reaches the final equilibrium state $\alpha = 1$. Symmetric considerations hold for the backward transition $1 \to 0$. The dimensionless vector fields $\mathbf{f}_\alpha^*(\mathbf{x})$ are depicted as arrows in each panel of Fig. 4, where, for the sake of better readability, they are plotted only on the $\phi = 0$ isoline. It should be noticed that the scale of the arrows changes from panel to panel, at least for the upper frames, $i = 11, \ldots, 14$. For the forward process, the latter are the configurations achieved just before the critical state. In this region, the MEP is particularly steep, requiring more intense forces, which result to be strongly localized near the vesicle contact region. On the contrary, the backward process requires a more distributed force field, as shown in images $i = 15, \ldots, 40$. The arrows reverse their direction between configurations $i = 14$ and $i = 15$, showing that in this interval the force field vanishes, confirming that the critical state occurs somewhere between these two images.

**Insights into the action of proteins from the MEP**. The forces required for overcoming the fusion topological barrier are stronger than those relative to fission, thus suggesting that the sole mechanical action of proteins may be complemented by additional features. For example, setting[10] $k = 20\, k_BT$, the resulting activation energy, $\Delta E_{0\to1}^\dagger \approx 226\, k_BT$, is associated with a very steep free energy profile. Consistently with the present findings, Deserno[72] suggests that fusion proteins, besides mechanical action, may contribute to lowering the energy barrier by locally modifying the Gaussian modulus in the contact region of the approaching membranes. Indeed, the introduction of a suitable, spatially dependent Gaussian modulus is expected to reduce the stiffness associated with the Gauss–Bonnet theorem, opening alternative routes to the topological change. Our results show that this scenario is actually possible since the forces associated with the Gaussian energy are localized in the region of contact between the two spheres and, therefore, it is reasonable that a variation of $k_G$ in such a region could lower the activation

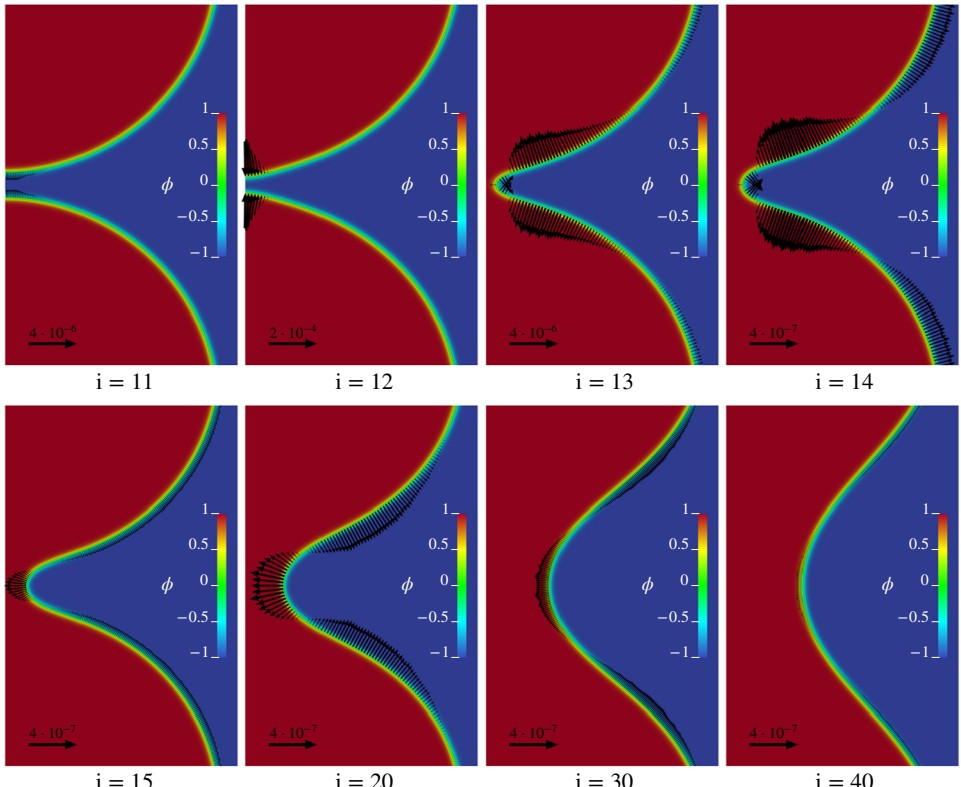

**Fig. 4 Force fields along the MEP.** Detailed views in the $r^\star - z^\star$ plane of the vesicle configurations. The index $i = (N-1)\alpha_i + 1$ numbers the images on the string. Vectors, that are plotted for clarity only on the $\phi = 0$ isoline, provide the force field $\mathbf{f}^\star$ required to keep the vesicle in equilibrium in the given configuration, balancing the internal elastic reaction. The contours depict the field $\phi$. For better visibility, vectors are scaled according to the reference arrow in each plot.

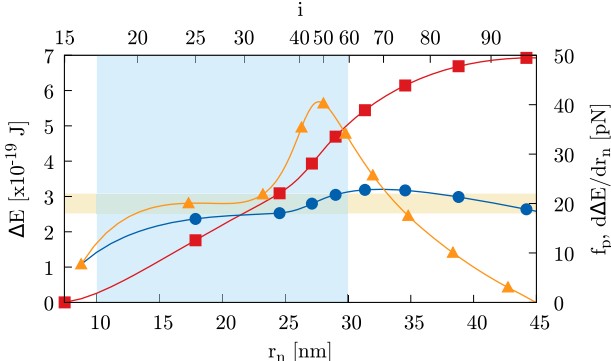

**Fig. 5 Proteins and constriction forces.** Red curve with squares: the energy needed to complete the fission as a function of the current neck radius $r_n$, $\Delta E = E(r_n) - E(r_0)$ vs. $r_n$. The second abscissa axis on top of the frame provides the image number $i$ along the MEP. Orange curve with triangles: estimated constriction force (second ordinate axis on the right), $d\Delta E/dr_n$ vs. $r_n$. Blue curve with dots: $f_p = \Delta E(r_n)/(r_n - r_0)$ vs. $r_n$. The vertical light blue band represents the range in which dynamin polymerizes[74]. The horizontal light orange strip depicts the value of dynamin constriction force measured in experiments[34, 74].

energy. For example, this situation is compatible with the observation that influenza virus hemagglutinin proteins, in addition to having apposition activity, are also able to perturb the membrane lipid bilayer by insertion of their amphipathic fusion peptide[73]. In this regard, the present phase-field approach can be easily adapted to the instance of a topological transition with a spatially dependent Gaussian modulus, a case we leave for future work.

As anticipated, the forces at play during fission are more distributed and less intense than for fusion. The large region they act on (Fig. 4), is consistent with the cooperation of several protein systems, like, e.g., in clathrin-mediated endocytosis, which involves clathrin polymerization and the subsequent action of the constrictase dynamin[36]. One can estimate the minimal work the protein system needs to perform to induce the topological change by comparing the free energy barrier $\Delta E_{1\to0}^\dagger$ with the protein work $\mathcal{W}_{1\to0} = f_p \Delta r$, where $f_p$ is the order of magnitude of the protein force and $\Delta r = r_{max} - r_0$ is the change in vesicle radius at the neck, between the equilibrium prolate ($r_{max}$) and the saddle point configurations ($r_0$). Given the scale of the system described above, we find $\Delta r = 37.4$ nm which, from the barrier height, provides $f_p = 0.91 \, k/(k_B T)$ pN. For the values of $k$ proper of fluid lipid membranes, we thus obtain protein forces in fairly good agreement with the experimental estimates[34], e.g. $\simeq 20$ pN for dynamin, $\simeq 65$ pN for ESCRT-III and $\simeq 80$ pN for FtsZ. For example, by assuming $k = 20 \, k_B T$, we obtain a protein constriction force $f_p$ of 18.2 pN. For the same bending rigidity, Fig. 5 shows —red curve with squares— the energy needed to complete the fission process as a function of the current neck radius $r_n$, $\Delta E(r_n) = E(r_n) - E(r_0)$ (note that the fission proceeds from larger to smaller neck radii, i.e. from right to left along the abscissa). The corresponding image number $i$ along the MEP is provided on the second abscissa axis on the top of the frame. The slope of the plot, $d\Delta E/dr_n$, orange line with triangles, provides the estimate of the constriction force (positive when constrictive). A plateau is apparent at $d\Delta E/dr_n \simeq 20$ pN in the range of radii $16 \leq r_n \leq 21$ nm. Notably, it is known from the literature[74] that, e.g., dynamin polymerizes on tubules with radius between 10 and 30 nm, exerting forces of the order of 20 pN. In order to facilitate

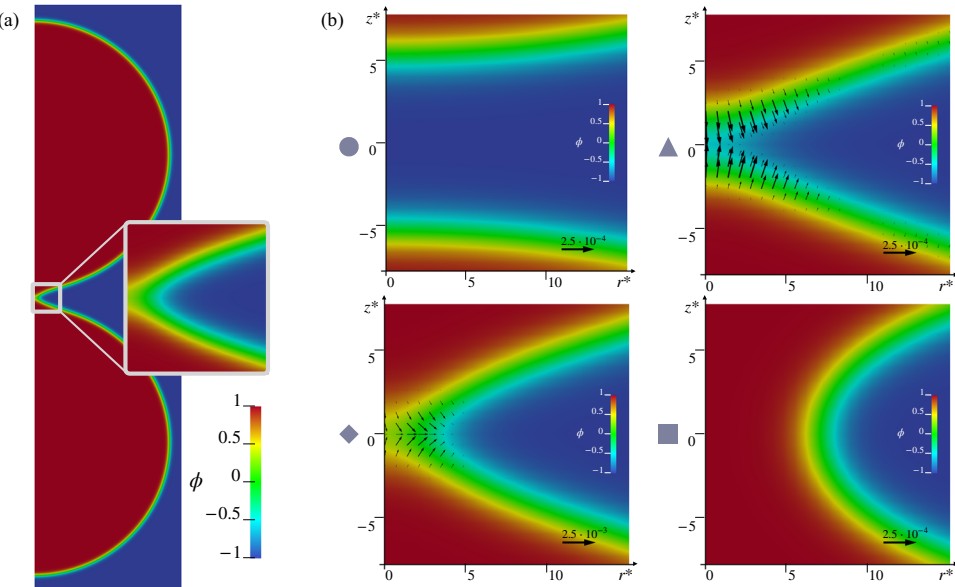

**Fig. 6 Close-ups of the merging region. a** Full-scale vesicle configuration highlighting the merging region enlarged in panel **b**. **b** Proceeding in the forward direction: equilibrium of the two bilayers (circle); merging of the proximal interface region (triangle); merging of the distal interface region (rhombus); saddle point configuration (square). Considering $\ell_{pf} = \ell_{me}$ as described above, the initial distance between the two vesicles is about 3.85 nm. Vectors depict the force field $\mathbf{f}^*$ required to keep the vesicle in equilibrium in the given configuration, balancing the internal elastic reaction. Vectors are scaled according to the reference arrows (note the tenfold increase in the bottom left plot of the panel).

comparison with published data, Fig. 5 also provides in blue, with dots, $f_p = \Delta E(r_n)/(r_n - r_0)$.

**Discussion**. We have provided a description of the full-scale process of topology change in the fusion/fission process of two large unilamellar vesicles (LUVs) with an approach that can be extended to deal with giant unilamellar vesicles (GUVs). The proposed free energy accounts for the Canham–Helfrich Gaussian energy jumps as prescribed by the Gauss–Bonnet theorem, and, far from the topological changes, recovers the Canham–Helfrich Hamiltonian itself in the limit of small bilayer thickness. However, during topological transitions, when the relative distance between approaching membrane segments becomes comparable to the bilayer thickness, the scale invariance of the asymptotic Canham–Helfrich Hamiltonian is broken. For such a reason, we defined the scale of our system by matching the lipid bilayer thickness with the diffuse interface width.

The smoothness of the free energy functional is the key aspect to featuring a well-defined chemical potential $\delta E/\delta\phi$, and, physically, a meaningful distributed force field also across the topological transition. In other words, the functional derivative of the Gaussian energy allows access to the force preventing the topological change, thus extending the realm of the classical, $k_G$-independent, shape equation, which, being obtained at constant topology, cannot account for the work done against the topological constraints. From a purely mathematical standpoint, our proposal should be interpreted as a rational way to regularize the singularity of the process and smoothly match the external solution before and after the transition, thus enabling the deployment of powerful variational approaches.

As anticipated in the section "Introduction", the results explicitly presented in the paper concern the simple, yet extremely important, case of the fusion of two identical LUVs with zero spontaneous curvature, together with the reverse process of fission. The approach is however readily applicable to more complex configurations, as shown by the few more cases illustrated in the Supplementary Discussion, where, in addition to numerical validation simulations, we report more data

concerning (i) the MEP for the transition between two spheres and a single one with no volume constraint, (ii) the fission of a prolate shape into two spheres induced by a spontaneous curvature, (iii) the forced fusion between two nested spheres to form a stomatocyte, and (iv) the forced transformation of an oblate vesicle ($g = 0$) into a torus ($g = 1$). Clearly, several microscopic effects are not included in the model. In any case, the impression is that the energetic correction due to such microphysics is small as compared to the energy barrier associated with the full-scale evolution of the vesicle.

It may be interesting to speculate whether more detailed information on the local structural rearrangement of the bilayers can be obtained from the present mesoscale model. For this reason, we extract enlarged views of the merging region, as depicted in Fig. 6a. The close-ups, shown in Fig. 6b, correspond to states between images $i = 10$ and 14, thus along the steepest stretch of the MEP of Fig. 2b (black line). These states have been obtained as a refinement of the original string. To this purpose, a new, finer string has been evolved with the final configuration fixed at the saddle point $i = 14$. The initial image of the new string, which is left free to fall down along the MEP, has found its equilibrium somewhere between images $i = 10$ and 11 in the original string. In the forward direction, the progression of the topological change is apparent (Fig. 6b), with the fusion of the outer parts of the contacting interfaces occurring first. The process is completed after the merging of the distal parts and the subsequent connection of the volumes originally enclosed by the disjoined vesicles. The vectors in each snapshot provide the distributed force field $\mathbf{f} = -\delta E/\delta\phi\nabla\phi$, namely the external force needed to counterbalance the elastic reaction, in the same spirit as Fig. 4. The external field pushes the two bilayers one against the other in the contact region until the proximal parts of the approaching interfaces get fused (top right). Once the distal parts of the interfaces start to get merged, the external force pushes outward (bottom left) in order to form a pore-like connection between the two vesicles (bottom right). At this moment the critical state is reached, and, from then onwards, the neck tends to spontaneously expand under the action of its own elasticity.

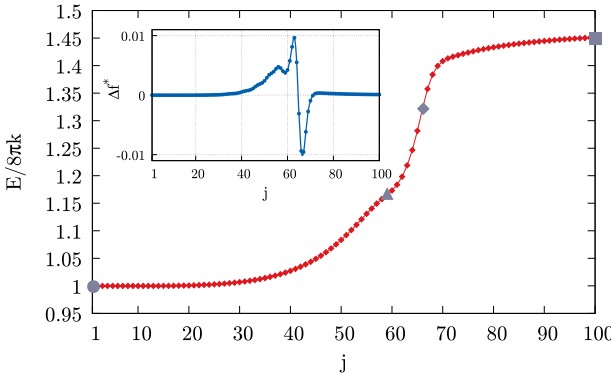

**Fig. 7 MEP steepest stretch zoom.** Refinement of the black line of Fig. 2b, obtained by evolving a new string made up of other 100 images, indexed with $j$ (main plot, red line with small rhombuses). Image $j = 100$ corresponds to $i = 14$ of the original MEP, while $j = 1$ lies somewhere between images $i = 10$ and $i = 11$. The gray symbols identify the four configurations of Fig. 6b: equilibrium of the two bilayers (circle); merging of the proximal interface region (triangle); merging of the distal interface region (rhombus); saddle point configuration (square). The blue line with small dots in the inset shows the difference between the forces on the $\phi < 0$ and $\phi > 0$ regions of a single interface as described in the main text. Note that the triangle and rhombus states identified on the main plot correspond to the two minima of the inset.

These intermediate configurations are reminiscent of those found in experiments[40] and MD simulations[22,23,28], i.e. pre-fusion states, stalks, hemifusions, and fusion pores. None of these is a metastable state in our case. Recent MD results with coarse-grained force fields show, however, that the stalk state, in particular, may become (meta)stable, decreasing the initial distance between the disjoined vesicles, itself related to the hydration state of the bilayers[24,27]. Clearly, hydration effects would need to be introduced through a suitable potential[75], in particular, to account for close apposition work expenditure. Figure 7 shows the zoom of the steepest stretch of the MEP obtained with the new 100 images of the finer string. The force fields in Fig. 6 can be exploited to evaluate the differential forces acting on the hypothetical leaflets

$$\Delta f = \int_D -\mathrm{sign}(\phi)\frac{\delta E}{\delta \phi}|\nabla\phi|\,dV, \qquad (9)$$

where $D$ denotes the upper half of the merging region in the plots of Fig. 6b, the sign of $\phi$ accounts for the difference between the forces across the dividing surface, and $-\delta E/\delta\phi|\nabla\phi|$ is the normal component of the (external) force density $\mathbf{f}$. $\Delta f$ is shown in the inset of Fig. 7. It is positive where the external force pulls the leaflets apart. Apparently, the differential of the force tends to lyse the bilayer during the first phase, after which the leaflets are packed back together when $\Delta f$ becomes negative. The two configurations denoted by a triangle and a rhombus in Fig. 6 correspond to the two (positive and negative, respectively) minima of the disjoining force.

Additional microscale effects could eventually be introduced via a local, configuration-dependent Gaussian modulus, taking inspiration from the expression $k_G = 2\left(k_G^{ml} - k^{ml} z_0 m^{ml}\right)$, which, in the context of the Canham–Helfrich theory[11], provides the Gaussian modulus of a bilayer. Here, $k_G^{ml}$, usually negative, is the Gaussian modulus, $k^{ml}$ the bending rigidity, and $m^{ml}$ the spontaneous curvature of the constituent (symmetric) monolayers of a given vesicle. $z_0$ can be interpreted as a measure of the thickness of the bilayer. The presence of the parameter $z_0$ in the classical theory can be exploited to introduce an additional, local

scale that modulates $k_G$ of the phase-field interface, particularly in the region where the lamellar structure of the bilayer is lost due to the topology rearrangement.

It may be noted that the above expression for the Gaussian modulus accounts for the fusogenic effect of a negative spontaneous curvature of the monolayer, $m^{ml} < 0$, which tends to reduce the absolute value of the bilayer Gaussian modulus, $|k_G|$, lowering the fusion barrier as shown in the inset of Fig. 2b. As already anticipated, the fission branch remains substantially unaltered. Consistently, the force field along the fission branch does not change appreciably with the Gaussian modulus. On the other hand, along the fusion branch, the forces are reduced by decreasing $|k_G|$, in line with the behavior of the corresponding barrier.

## Conclusions

Naively, one may argue that protein systems could have evolved in Nature to overcome the large barrier that stabilizes the vesicle topology by following a minimal energy pathway. Hence, by means of the proposed free energy functional, we have evaluated the minimal free energy path for the transition and extracted the force field able to drive the process with minimal work expenditure. The free energy profile we find shows the strong asymmetry between the fusion and the fission processes. For fusion, the required force field is extremely intense and suggests that proteins could locally modify the Gaussian modulus during the topological change, a case that can finally be addressed with the presented approach. On the contrary, as regards fission, the obtained spatial scales and forces are consistent with the experimental estimates for typical fission protein systems, like the ESCRT-III, FtsZ, and dynamin.

It may be stressed that the approach proposed here is readily extended to fully 3D configurations, see e.g. the "Numerical validation" section in the Supplementary Methods where the axisymmetric configuration of the two spheres was actually computed with a fully 3D simulation. This extension can be important in describing asymmetric neck geometries and their effects on the energy barrier[76,77].

To conclude, it can be noted that the proposed approach can naturally be coupled with hydrodynamics[78,79] to include the dynamics of external and internal aqueous environments. One may also observe that the Gaussian energy functional can find a much broader scope, e.g., as an indicator of the topological genus in the context of cluster analysis[80,81], or as a way to provide a barrier towards undesired/unphysical fusion processes. A compelling example concerns emulsions where surfactant-covered droplets behave much like lipid micelles[82,83], suggesting that the Gaussian energy could play a role in the emulsification process.

## Methods

**Sharp interface limit**. An energy $E[\phi]$ (Eq. (3)), is associated with each field configuration and is such as to admit local minimizers of the form

$$\phi(\mathbf{x}) = f\left(\frac{d(\mathbf{x})}{\epsilon}\right), \qquad (10)$$

where $d(\cdot)$ is the signed distance function from the membrane mid-surface $\Gamma$. We choose to define the signed distance such that $\mathbf{n} = \nabla d$ computed on $\Gamma$ is equal to the inward-pointing unit normal to the vesicle. Setting $d^*(\mathbf{x}) = d(\mathbf{x})/\epsilon$, we also require that $\lim_{d^* \to \pm\infty} \phi = \pm 1$ and $\phi = 0$ for $d = 0$. Therefore, $\pm 1$ are the values for the stable phases of the inside and outside bulk, and the level set $\phi = 0$ identifies the membrane mid-surface. Physically, the free energy functional should recover the Canham–Helfrich Hamiltonian (Eq. (1)), in the limit of a small width-to-vesicle-extension ratio. $E_B[\phi]$ models the bending energy of the membrane[44], while $E_G[\phi]$ is the term proposed here to account for the Gaussian energy.

As anticipated, our purpose here is to show that, under the general ansatz (10) and in the sharp-interface limit ($\epsilon/D_{ve} = \lambda \ll 1$), minimizing the phase-field free energy functional (3) is equivalent to minimizing the Canham–Helfrich free

energy. Denoting with a prime the derivative done with respect to $d^*(\mathbf{x})$, a direct computation leads to

$$E_B[\phi] = k\,\frac{3}{4\sqrt{2}}\,\lambda \int_{\bar\Omega}\left[\frac{1}{\lambda^2}\left(f'' - (f^2-1)f\right)\right.$$
$$\left. + \frac{1}{\lambda}\left(f'\bar\nabla\cdot\mathbf{n} + (1-f^2)\sqrt{2}\bar m\right)\right]^2 d\bar V, \quad (11)$$

$$E_G[\phi] = k_G\,\frac{35}{16\sqrt{2}}\int_{\bar\Omega}\frac{f'^4}{\lambda}\left[(\bar\nabla\cdot\mathbf{n})^2 + \mathbf{n}\cdot\bar\nabla(\bar\nabla\cdot\mathbf{n})\right] d\bar V, \quad (12)$$

where we have denoted with a bar the dimensionless lengths obtained by dividing by $D_{ve}$. Therefore, in order to minimize $E = E_B + E_G$, as $\lambda \to 0$, the leading-order term $f_0$ of $\phi(\mathbf{x}) = f(d^*(\mathbf{x})) = f_0(d^*(\mathbf{x})) + \sum_{i=1}^{+\infty}\lambda^i f_i(d^*(\mathbf{x}))$ must satisfy $f_0'' = (f_0^2 - 1)f_0$, which has the solution

$$f_0(d^*(\mathbf{x})) = \tanh\left(\frac{d(\mathbf{x})}{\epsilon\sqrt{2}}\right). \quad (13)$$

Hence, $\epsilon$ is actually related to the width of the interface. Moreover, by repeating the computations done by Wang[84] for the bending energy alone, it is possible to show that, also in the presence of the Gaussian energy term, one finds $f_1(d^*(\mathbf{x})) \equiv 0$ (see the Supplementary Methods for the whole computation). Therefore, given that $\sqrt{2}f_0' = (1 - f_0^2)$, we are left with

$$E_B[\phi] = k\,\frac{3}{4\sqrt{2}}\int_{\bar\Omega}\frac{f_0'^2}{\lambda}\left(\bar\nabla\cdot\mathbf{n} + 2\bar m\right)^2 d\bar V + O(\lambda), \quad (14)$$

$$E_G[\phi] = k_G\,\frac{35}{16\sqrt{2}}\int_{\bar\Omega}\frac{f_0'^4}{\lambda}\left[(\bar\nabla\cdot\mathbf{n})^2 + \mathbf{n}\cdot\bar\nabla(\bar\nabla\cdot\mathbf{n})\right]d\bar V + O(\lambda^2). \quad (15)$$

Denoting with $k_1$ and $k_2$ the principal curvatures, we have $\nabla\cdot\mathbf{n} = -(k_1+k_2) = -2M$ and $\mathbf{n}\cdot\nabla k_i = k_i^2$, with the result that $(\nabla\cdot\mathbf{n})^2 + \mathbf{n}\cdot\nabla(\nabla\cdot\mathbf{n}) = 2k_1k_2 = 2G$. Now, noticing that for $\lambda\to 0$ one finds $f_0'^2(\bar d(\mathbf{x})/\lambda)/\lambda \xrightarrow{\mathcal{W}} 2\sqrt{2}/3\,\delta(\bar d(\mathbf{x}))$, $f_0'^4(\bar d(\mathbf{x})/\lambda)/\lambda \xrightarrow{\mathcal{W}} 8\sqrt{2}/35\,\delta(\bar d(\mathbf{x}))$, where $\delta(x)$ is the Dirac delta function and $\mathcal{W}$ denotes a weak limit in the sense of distributions, and getting back to dimensional variables, the asymptotic behavior follows as

$$E[\phi] \sim 2k\int_\Gamma (M-m)^2\,dS + k_G\int_\Gamma G\,dS, \quad (16)$$

i.e., the phase-field energy functional reproduces the Canham–Helfrich free energy in the sharp-interface limit ($\epsilon/D_{ve} \ll 1$). It is worth noticing that the inclusion of the Gaussian energy, which is subdominant in $\lambda$, preserves the hyperbolic tangent form (13) of the leading order solution together with $f_1(d^*(\mathbf{x})) \equiv 0$, as for the more standard model with the bending energy alone[44]. Since $f_1(d^*(\mathbf{x})) \equiv 0$, the desired expression of the bending energy is retained at order $\lambda^{-1}$, and the accuracies $O(\lambda)$ and $O(\lambda^2)$ are guaranteed in Eqs. (14) and (15), respectively. Furthermore, in our formulation, the phase-field Gaussian energy (6) has no singularities and actually depends at most on derivatives of order two, as it is possible to see by replacing $\nabla\phi\cdot\nabla\nabla^2\phi$ with $\nabla^2|\nabla\phi|^2/2 - \mathbf{H}_\phi : \mathbf{H}_\phi$ in Eq. (7), where $\mathbf{H}_\phi$ is the Hessian matrix of the field. As regards the incompressibility of the membrane, we impose the geometrical constraints described in the section "Results and discussion" using the functionals

$$A[\phi] = \frac{3}{4\sqrt{2}}\epsilon\int_\Omega\left[\frac{(1-\phi^2)^2}{2\epsilon^2} + |\nabla\phi|^2\right]dV, \quad (17)$$

$$V[\phi] = \int_\Omega\frac{(1+\phi)}{2}\,dV, \quad (18)$$

which, respectively, behave like the vesicle area and enclosed volume in the sharp interface limit.

**Gauss–Bonnet theorem.** Let us assume that

$$\phi(\mathbf{x}) = \tanh\left(\frac{d(\mathbf{x})}{\epsilon\sqrt{2}}\right), \quad (19)$$

where $\mathbf{x}\in\Omega$, being $\Omega$ a cylindrical domain of radius $R$ and height $L$ in the ordinary three-dimensional space, and $d(\cdot)$ the signed distance from an axisymmetric surface in $\Omega$. This assumption leads to $|\nabla\phi| = (1-\phi^2)/(\epsilon\sqrt{2})$, and, moreover, we set $h(\phi) = \left[(1-\phi^2)/(\epsilon\sqrt{2})\right]^4$. Using the cylindrical coordinates system, it is possible to show by a direct computation[56] that one of the two principal curvatures is $k_1 = -\partial_r\phi/(r|\nabla\phi|)$. Therefore, remembering that $\nabla\cdot\mathbf{n} = -(k_1+k_2)$ and

**Table 1 Gaussian energy computed values, $k_G = -k$.**

| Shape | $E_G/8\pi k$ (exact) | $E_G/8\pi k$ (numerical) |
|---|---|---|
| Sphere | $-5.0\times 10^{-1}$ | $-5.000525\times 10^{-1}$ |
| Torus | 0 | $-1.729446\times 10^{-18}$ |
| Cylinder | 0 | $-9.860761\times 10^{-32}$ |

$\mathbf{n}\cdot\nabla k_i = k_i^2$, with $\mathbf{n} = \nabla d$, Eq. (15) can be rewritten as

$$E_G[\phi] = k_G\,\frac{35}{8\sqrt{2}}\epsilon^3\int_\Omega h(\phi)\,k_1 k_2\,dV$$
$$= -k_G\,\frac{35}{8\sqrt{2}}\epsilon^3\int_\Omega h(\phi)\,\nabla\cdot(\mathbf{n}\,k_1)\,dV$$
$$= k_G\,\frac{35}{8\sqrt{2}}\epsilon^3\int_\Omega\frac{dh}{d\phi}\,\nabla\phi\cdot\mathbf{n}\,k_1\,dV + I_{\partial\Omega}$$
$$= k_G\,\frac{35}{8\sqrt{2}}\epsilon^3\int_\Omega\frac{dh}{d\phi}\,|\nabla\phi|\,k_1\,dV + I_{\partial\Omega}$$
$$= -k_G\,\frac{35}{4\sqrt{2}}\epsilon^3\pi\int_{-L/2}^{+L/2}dz\int_0^R\frac{dh}{d\phi}\frac{\partial\phi}{\partial r}\,dr + I_{\partial\Omega}$$
$$= k_G\,\frac{35}{4\sqrt{2}}\epsilon^3\pi\int_{-L/2}^{+L/2}[h(\phi(r=0,z)) - h(\phi(r=R,z))]dz + I_{\partial\Omega},$$

where

$$I_{\partial\Omega} = -k_G\,\frac{35}{8\sqrt{2}}\epsilon^3\int_{\partial\Omega}h(\phi)\,k_1\,\mathbf{n}_\Omega\cdot\mathbf{n}\,dS.$$

Supposing to have a single, connected, closed surface, after letting $\Omega$ invade $\mathbb{R}^3$, and considering relation $f_0'^4(\bar d(\mathbf{x})/\lambda)/\lambda \xrightarrow{\mathcal{W}} 8\sqrt{2}/35\,\delta(\bar d(\mathbf{x}))$ applied to $h(\phi)$, we obtain

$$\lim_{\epsilon\to 0}E_G[\phi] = 2\pi k_G\int_{-\infty}^{+\infty}\delta(d(r=0,z))\,dz$$
$$= 4\pi k_G\,(1-g),$$

recovering the Gauss–Bonnet theorem (Eq. (2)), in the axially symmetric case. The last equality is justified by the fact that the Dirac delta function counts the intersections of the surface with the $z$-axis, which is equivalent to checking whether the surface has a hole.

**Numerical scheme.** The numerics relies on FFT-based spectral differentiation in cell-centered grids which provide high-accuracy solutions, with special regard to the estimate of the Gaussian energy. The accuracy in evaluating the Gaussian energy (Eq. (6)), is shown in Table 1 for a sphere, a torus, and a straight cylinder. Given the axial symmetry of these shapes, all the computations are done in a $[0, 40]\times[0, 40]$ computational domain in the $r^* - z^*$ plane with a grid of $80\times 80$ nodes. In evaluating the functional, we set $\phi(\mathbf{x}^*) = \tanh((\sqrt{r^{*2} + (z^* - 20)^2} - 10)/\sqrt{2})$ for the sphere, $\phi(\mathbf{x}^*) = \tanh((\sqrt{(r^* - 20)^2 + (z^* - 20)^2} - 10)/\sqrt{2})$ for the torus and $\phi(\mathbf{x}^*) = \tanh((r^* - 10)/\sqrt{2})$ for the cylinder, which is obtained by imposing Eq. (13).

The energy pathways of the section "Results and discussion" and the MEP with unconstrained volume in the Supplementary Discussion are obtained by means of the string method, which is briefly described below. The remaining simulations reported in this paper, i.e. the one shown in Fig. 1 and those in the Supplementary Discussion and Supplementary Methods, are carried out using the Allen–Cahn dynamics

$$\frac{\partial\phi}{\partial t} = -M\frac{\delta\bar E}{\delta\phi}, \quad (20)$$

where $M$ is the mobility coefficient and $\delta\bar E/\delta\phi$ is the functional derivative of the augmented energy

$$\bar E[\phi] = E[\phi] + \gamma(A[\phi] - A_0) + \frac{1}{2}M_1(A[\phi] - A_0)^2$$
$$+ \Delta p(V[\phi] - V_0) + \frac{1}{2}M_2(V[\phi] - V_0)^2. \quad (21)$$

Here, the additional terms added to the energy (3) are needed when constraining to $A_0$ and $V_0$ the vesicle area (17) and volume (18), respectively. $M_1$, $M_2$ are two penalty constants, whereas $\gamma$ and $\Delta p$ are updated at each time step according to the augmented Lagrangian method[85]:

$$\gamma^{n+1} = \gamma^n + M_1(A[\phi^{n+1}] - A_0), \quad (22)$$

$$\Delta p^{n+1} = \Delta p^n + M_2(V[\phi^{n+1}] - V_0). \quad (23)$$

Therefore $\gamma$ and $\Delta p$ are estimates of the Lagrange multipliers that improve at every time step. Starting from an assigned initial condition, the Allen–Cahn

dynamics causes the energy to monotonically decrease in time until it reaches a critical steady-state. The dimensionless time and mobility are $t^* = t/\tau_R$ and $M^* = 8\pi k M \tau_R/\epsilon^3$, respectively, with $\tau_R$ a suitable time scale.

With the help of the PETSc library[86], a Crank–Nicolson time-stepping scheme is employed to integrate the Allen–Cahn gradient flow, while a semi-implicit Euler single-step scheme is used to solve the more computationally demanding string dynamics. The explicit form of the functional derivative $\delta \bar{E}/\delta\phi$ is given in the Supplementary Methods together with some numerical experiments carried out to validate the approach.

**String method**. The zero-temperature string method[57] is a technique for computing free energy barriers and transition pathways on a given energy landscape. The method proceeds by evolving in time a *string*, namely a curve parameterized by $\alpha \in [0, 1]$. For each $\alpha$ the image of the string is a phase-field function $\phi_\alpha(\mathbf{x})$ representing a membrane state.

Given an initial guess for the pathway connecting two local minima, the string evolves in time following the dynamics

$$\frac{\partial \phi_\alpha}{\partial t} = -M \left( \frac{\delta \bar{E}}{\delta \phi_\alpha} \right)^\perp \quad \forall \alpha \in [0, 1], \tag{24}$$

where $M$ is a mobility coefficient, $\delta \bar{E}/\delta \phi_\alpha$ is the functional derivative of Eq. (21) evaluated on the image $\phi_\alpha$ and $(\delta \bar{E}/\delta \phi_\alpha)^\perp$ is its component normal to the string. This last quantity can be computed as $(\delta \bar{E}/\delta \phi_\alpha)^\perp = \delta \bar{E}/\delta \phi_\alpha - \langle \delta \bar{E}/\delta \phi_\alpha | \tau \rangle \tau$, where $\tau = \partial_\alpha \phi_\alpha / \langle \partial_\alpha \phi_\alpha | \partial_\alpha \phi_\alpha \rangle^{1/2}$ is the unit tangent to the string and $\langle \cdot | \cdot \rangle$ is the standard $L_2$ inner product. In this way, at a steady state, the string converges to a minimal energy path[65]. In order to eliminate the trouble of projecting the functional derivative and in order to use the equal arc-length parameterization, the string dynamics can be rewritten[67] as

$$\frac{\partial \phi_\alpha}{\partial t} = -M \frac{\delta \bar{E}}{\delta \phi_\alpha} + \bar{\lambda} \tau \quad \forall \alpha \in [0, 1], \tag{25}$$

where $\bar{\lambda} = \lambda + M \langle \delta \bar{E}/\delta \phi_\alpha | \tau \rangle$ and $\lambda$ is a Lagrange multiplier for the purpose of enforcing the chosen parameterization $\partial_\alpha \langle \partial_\alpha \phi_\alpha | \partial_\alpha \phi_\alpha \rangle^{1/2} = 0$.

The algorithm follows the steps:

1. Evolution from $t$ to $t + \Delta t$ of the discrete string, made up of $N$ images $\phi_i$, with the dynamics

$$\frac{\partial \phi_i}{\partial t} = -M \frac{\delta \bar{E}}{\delta \phi_i}, \quad i = 1, \dots, N.$$

Time integration is performed in wave number space by means of the semi-implicit Euler single-step scheme. The evolved images at time $t + \Delta t$ are denoted as $\tilde{\phi}_i$.

2. Computation of the arc lengths corresponding to the evolved images:

$$s_1 = 0,$$
$$s_i = s_{i-1} + \langle \tilde{\phi}_i - \tilde{\phi}_{i-1} | \tilde{\phi}_i - \tilde{\phi}_{i-1} \rangle^{1/2},$$
$$i = 2, \dots, N.$$

Thus, the evolved images have parameters $\alpha_i = s_i/s_N$.

3. Linear interpolation of the evolved images in order to compute the new images at equal arcs $\alpha_i = (i-1)/(N-1)$. These are the actual solutions at time $t + \Delta t$. It is worth noticing that linear interpolation conserves vesicle volume.

4. Go back to one and iterate until convergence.

**Force fields computation**. Given a membrane state, it is possible to compute the external force needed to balance the elastic force arising from the energy of the membrane. For this purpose, let us consider an arbitrary and infinitesimal variation $\delta\phi$ of the phase field, consistent with the area and volume constraints if present. This variation results in a spatial displacement $\delta\mathbf{x}$ of the field lines. The displacement can be thought to occur in a virtual time interval $\delta t$, within which the field lines move with a virtual velocity $\mathbf{u}$ such that $\partial\phi/\partial t = -\nabla\phi \cdot \mathbf{u}$ (null material derivative condition). By integrating in time this last equation from $t$ to $t + \delta t$, we are left with the first-order approximation

$$\delta\phi = -\nabla\phi \cdot \mathbf{u}\delta t = -\nabla\phi \cdot \delta\mathbf{x}. \tag{26}$$

Hence, the work performed by the external force field $\mathbf{f}$ to deform the membrane is

$$\int_\Omega \mathbf{f} \cdot \delta\mathbf{x}\, dV = \delta\bar{E} =$$
$$\int_\Omega \frac{\delta\bar{E}}{\delta\phi} \delta\phi\, dV = -\int_\Omega \frac{\delta\bar{E}}{\delta\phi} \nabla\phi \cdot \delta\mathbf{x}\, dV, \tag{27}$$

and one can identify the force field

$$\mathbf{f} = -\frac{\delta\bar{E}}{\delta\phi}\nabla\phi, \tag{28}$$

thanks to the arbitrariness of $\delta\mathbf{x}$.

## Data availability

The datasets generated during and analyzed during the current study are available from the corresponding author on reasonable request.

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

## Acknowledgements

Support is acknowledged from the 2020 Sapienza Large Project: Dynamics of Biological and Artificial Lipid Bilayer Membranes. Concerning computational resources, we acknowledge PRACE for awarding us access to Marconi's successor at CINECA, Italy, PRACE 23rd call project No. 2021240074; DECI 17 SOLID project for resource Navigator based in Portugal at https://www.uc.pt/lca/from the PRACE aisbl; CINECA award under the ISCRA initiative, for the availability of high-performance computing resources and support (ISCRA-B FHDAS, ISCRA-C ToTraVes).

## Author contributions

C.M.C. designed the study; M.Bo. contributed the Gaussian free energy model; M.Bo., M.Bu. designed the simulation campaign under the supervision of M.G.; M.Bo. performed the simulations; M.Bu. processed the data; M.Bo., M.G., M.Bu., C.M.C., analyzed the simulation data, discussed the results, contributed writing, revised, and approved the final version.

## Competing interests

The authors declare no competing interests.
