## [Peer Review File · Communications Physics]

Reviewers' comments:

Reviewer #1 (Remarks to the Author):

Phase field methods are well established as alternative descriptions of physical phenomena related to interfaces. Contrary to mathematical treatments where interfaces are described as infinitely thin surfaces immersed in a space of higher dimensions, phase-fields are described everywhere in the host space and the interface is associated with a value or a range of values of the field, typically around its zero point. Pursuing the development of phase-field tools to describe fluid lipid membranes, Bottacchiari et al complete in this paper the standard elastic description of the membranes within the framework of phase-field methods by adding the so-called Gaussian contribution to the energy. Importantly, such addition allows to describe topological transitions in membrane systems, where the Gaussian terms of the energy play a preponderant role.

As a test-case the authors describe the topological transformation occurring between two spheres and one sphere: vesicle fusion in the forward direction of the transformation and vesicle fission in the backward direction. The elastic contributions to the shape and topology transformations are followed along a minimal energy path, thus allowing to point to the main characteristics of the energy barrier that separated the two vesicular states. The method also allows to study the distribution of the forces acting on the interface as the transformation proceeds and thus to speculate on how these forces might or might not be induced by typical proteins involved in the biologically relevant fusion and fission processes.

The paper is very well written, the goals clearly stated and the methods introduced and developed here are presented in a rigorous yet straightforward manner. I did not detect any technical inconsistency either in the main text or in the supplementary material. I do challenge however several assertions of the authors and will consider recommending the paper for publication pending their responses to my comments and criticism.

Perhaps the most debatable idea in the paper is that the phase-field method dramatically improves the "reality" of the description of the topological transition over other approaches. Molecular dynamic simulation show that topological transitions initiate at the molecular level with rearrangements of the lipid leaflets that cannot not be easily described by coarse-grained approaches, phase-field techniques included. It then proceeds by relaxation at increasingly larger scales, at the top of which one finds membrane shape evolutions likely to be well described by the Canham-Helfrich Hamiltonian or by its related formulations. There is thus a lower cutoff length at which all of these theories cease to work, either because of the non-linear character imposed by strong curvatures or other factors such as strong in-plane stress and concentration gradients and so on. Incidentally Fig. 3 comforts the idea that the standard Canham-Helfrich description would also provide a rather accurate description of the topological transition considered here. It seems to this referee that the phase field method is indeed convenient as no cuts need to be introduced in the surfaces during fission or fusion, but does not fundamentally describe the actual physical process in a more realistic manner.

How "natural" or realistic is the phase-field method should also be assessed against the actual behavior of two vesicles pushed together: in most situations they repel and flatten at the contact point rather than fuse. Does then this method then describes membranes that always fuse and never repel each-other? If this is the case, what are the fundamental forces missing from the picture

and how could they be re-introduced?

Although the forces acting on the $\phi=0$ surface present some interest for the description of the shape evolution, could the differential of forces acting on one side or the other of the dividing surface bring additional information helping to bridge the gap towards the molecular scales? It would be for instance interesting to know if such an indicator or other related ones can provide an upper cutoff for the forces needed to lyse the membrane and initiate a merging stalk.

It is well known that lipids with reverse spontaneous curvature have fusogenic properties, how well can this approach account for non-zero spontaneous curvature?

Two last comments:

- The inset in Fig. 2 (c) perturbs the understanding of the information in the figure, without much added value.
- Fig. 4 is not optimal. Its color scheme in the grey part is confined to a very small region (the interface) and thus it does not convey any easily visible information. Since the force field is obviously symmetric around the midplane, one can confine it to the lower part of the figure and use a vector representation in the upper part for the normal, this would be more informative than the color scheme.

Reviewer #2 (Remarks to the Author):

The manuscript describes the fission and fusion of vesicles using Ginzburg-Landau type free energy. The large body of work focuses on a single example describing the fusion process of two spherical vesicles. Both vesicles have zero spontaneous curvature and after fusion, the final relaxed state is prolate. The presented results in Figure 4 are very interesting, as it illustrates the spatial force acting on the membrane before and after the fusion process, which aligns with one's intuition. The outcomes of the study are rather interesting and the methodology is quite novel in the field, but the study fails to address so many aspects of topological transitions. The report describes two genus-zero $g=0$ vesicles' topological transformation to form one genus-zero $g=0$ vesicle. However, from the general title of the paper one would expect other examples such as $g=1$ to $g=0$ transformation and higher genus vesicles, as an example please see the preprint by the Jülicher group which reports topological morphogenesis of organoids:

Ishihara, K., Mukherjee, A., Gromberg, E., Brugués, J., Tanaka, E. M., & Jülicher, F. (2021). Topological morphogenesis of neuroepithelial organoids. *bioRxiv*.

Despite the novelty of the method used in the context of membrane biophysics as well as interesting results, I believe the results would not interest a wide range of audiences. The general mathematical formalism seems rather complex compared to the simpler and widely used Helfrich model. One of the good features of the Helfrich model is the simplicity of the model not only for theoretical and computational adaptation but also for experimental studies. For instance, one can extract several essential parameters from the mean curvature of vesicles using a spherical cap approximation. Thus I doubt that the results of the study are simply reconcilable with the experimental investigations in the future. However, to strengthen the conclusion of the study I would encourage the authors to

address and clarify a few points about the study and expand the study to a larger parameter space as mentioned in the following.

1-One clear aspect is that a similar fusion process is not available for any arbitrary reduced volume simply due to geometrical constraints. It would be very useful if the authors can provide a supplementary figure or note discussing this geometrical aspect.

2-The Ginzburg-Landau phase-field employed in this study seems rather arbitrary and it is not clear how much the outcomes would change by changing the expression of the phase field. An elaboration regarding this issue is required.

3-The claim made in the second column of page five "...fusion and fission processes cannot take place spontaneously and require further agents to happen, in addition to the elasticity and thermal fluctuation" does not seem accurate. Reference 25 is a contradictory example of this claim, where membrane-bound proteins only induced positive spontaneous curvature which can cause the division of a dumbbell shaped vesicle into two spherical vesicles. Such spontaneous curvature can be induced by exposing the vesicle to other forms of asymmetries.

4-The effect of spontaneous curvature is overlooked in the large part of the study; yet, spontaneous curvature is one of the most important material parameters that can induce fission and fusion. Also, the initial reduced volume of the vesicle is not defined for most cases.

5-The gaussian modulus is assumed to be equal to the bending rigidity. This seems a plausible assumption. However, it would be worth showing how the energy landscape and force field of the fission and fusion changes if the gaussian modulus is larger or smaller than the bending rigidity.

6-The axial symmetry is assumed in the process. The effect of non-axially symmetric neck geometries is completely omitted. However, recent studies show that neck symmetry can break under different circumstances, for example, see.

Vasan, R., Rudraraju, S., Akamatsu, M., Garikipati, K., & Rangamani, P. (2020). A mechanical model reveals that non-axisymmetric buckling lowers the energy barrier associated with membrane neck constriction. *Soft Matter*, 16(3), 784-797.

Satarifard, V., Grafmüller, A., & Lipowsky, R. (2018). Nanodroplets at membranes create tight-lipped membrane necks via negative line tension. *ACS nano*, 12(12), 12424-12435.

7-The intermediate state of the membrane fusion is characterized by the formation of the Hemi-fused state. From the presented results it was not clear if the phase-field model can capture the existence of such a state.

8- One interesting morphology would be two nested vesicles where fusion would lead to stomatocyte morphology. In addition, a study of the topological transformation of spherical vesicle into toroidal vesicle would be very interesting.

Few final questions:

1-In the fission example (Figure 1), why do we only observe symmetric divisions?

- 2-What is the effect of reduced volume on the fission-fusion force field?
- 3-Did you study the hysteresis? How does the reversible pathway look like?
- 4-What would be the effect of non-zero temperature in the results of your study?
- 5-How does the computational cost of your method compare to the computation of the Helfrich model by minimizing triangulated membrane or solving ODEs?

Reviewer #3 (Remarks to the Author):

Bottacchiari and colleagues describe a mathematical framework to analyze the shapes of biological membranes. The present approach is an extension of a previous method that allows for topological transformation in membrane shape remodelling process. The author tests the framework on several systems and reports similar results to those reported in the literature. Moreover, in a certain limit (sharp interface limit), the equation transforms into the widely used Helfrich Hamiltonian. They also derive force fields and use it to obtain interesting analysis of different membrane configurations. Although the authors have defined an interesting and important problem, I do not see how the current work has addressed the problem, or whether the manuscript is sufficiently clear to indicate this. Therefore, I cannot recommend the current format for publication but I would like to see a revised manuscript and responses to questions and suggestions noted below.

Major comments/suggestions

An important aspect of the manuscript is a mathematical framework for calculating the minimum bending energy of a surface given a global surface area and volume limit. What is the purpose of focusing on the biomembranes and fusion process? Clearly, this creates a number of problems (see below). Compared to the Canham-Helfrich Hamiltonian (CHH), this approach also does not provide a correct energy landscape of the fusion process. For example, see S. Kawamoto et al., Free energy analysis along the stalk mechanism of membrane fusion. *Soft Matter*, 2014,10, 3048-3054 and C. S. Poojari et al, Free energies of membrane stalk formation from a lipidomics perspective *Nature Communications* 2, 6594 (2021). However, I see that the present study has "implications" for biomembrane shape, but provides no additional "reliable" information compared to CHH.

Please elaborate on the rationale behind equations 5 and 7, particularly 7 since this is a term introduced in this work? Are these terms unique functional forms, or could there be other forms that give the same results? Is the field a measure of membrane density? Does it have any physical meaning? Would you have any suggestions as to how one can extend (fine tune) this term based on, for example, simulation data to capture the correct energy landscape? Also, does the method provide a unique numerical answer if there is no constraint on vesicle volume? Membranes are permeable to water, so not in all conditions, volume is constant.

Topological transformation using Canham-Helfrich hamiltonian (CHH), can be captured through sudden moves, like the one presented in Gompper, G., and Kroll, D. M. (1998). *Phys. Rev. Lett.* 81, 2284–2287. The current work claims to have provided a smooth transition. However, in figure 2b, the number of the points in the transition region is only 4, and they are far (energetically) from each other.

On page 5, the authors write: $D=206$ nm is within the range of CHH validity, and well beyond the current limits of atomistic approaches. CHH becomes problematic when we deal with regions of high curvature; radii comparable to the membrane thickness. So, during membrane deformation, such regions could form even for extremely large vesicles (as it happens here too). Also, as to the second part of the statement: See for example below articles for current state of molecular dynamics simulations. They are not well below $D=206$ nm.

Pezeshkian, et al, <https://doi.org/10.1101/2021.09.15.459697>; Pezeshkian, et al, Backmapping triangulated surfaces to coarse-grained membrane models; *Nature Communications* 11, 2296 (2020); J. V. Vermaas, Assembly and Analysis of Cell-Scale Membrane Envelopes *J. Chem. Inf. Model.* 2022, 3, 602–617

Shape fluctuations (undulations) are an important feature of biomembranes (see for example L.Johannes, et al, Clustering on Membranes: Fluctuations and More *Trends Cell Biol.*, 28 405-415 (2018)). What does equation 3 transform to for a flat membrane with small deviations? Does it capture the correct undulations spectrum?

Minor comments/suggestions

CHH works for length scales much smaller than the one reported in the current manuscript: see Fiorin G, Direct derivation of free energies of membrane deformation and other solvent density variations from enhanced sampling molecular dynamics. *J Comput Chem* 2020, 41:449–459,

On page two, a sentence with “most of the current understanding is from the experiment”, contradicts the previous paragraph and is also not accurate.

At the end of page 2, what does topological information mean?

Numerous applications, what are these applications?

Membrane tension associated with area change (there are other types of membrane tension) is much greater than the bending energy. Is this not a better argument for a constant area algorithm?

**Reply to the comments of
Referee 1
to the paper
Topological transitions in fluid lipid vesicles:
activation energy and force fields**

July 30, 2022

We thank the Referee for their words of overall appreciation of our work,

“The paper is very well written, the goals clearly stated and the methods introduced and developed here are presented in a rigorous yet straightforward manner. I did not detect any technical inconsistency either in the main text or in the supplementary material.”

“Importantly, [the addition of the Gaussian term to the energy] allows to describe topological transitions in membrane systems, where the Gaussian terms of the energy play a preponderant role.”

“I ... will consider recommending the paper for publication pending their responses to my comments and criticism.”

We eagerly meet the challenge of better explaining all the assertions that the Referee pointed out in their report and accordingly modify the paper. In doing so, we report the original questions in italics followed by our answers.

- *“Perhaps the most debatable idea in the paper is that the phase-field method dramatically improves the “reality” of the description of the topological transition over other approaches. Molecular dynamic simulation show that topological transitions initiate at the molecular level with rearrangements of the lipid leaflets that cannot not be easily described by coarse-grained approaches, phase-field techniques included. It then proceeds by relaxation at increasingly larger scales, at the top of which one finds membrane shape evolutions likely to be well described by the Canham-Helfrich Hamiltonian or by its related formulations. There is thus a lower cutoff length at which all of these theories cease to work, either because of the non-linear character imposed by strong curvatures or other factors such as strong in-plane stress and concentration gradients and so on.”*

We overall agree with the Referee’s comment but would like to better explain our position. Our original intent was simply to extend the Canham-Helfrich approach to bridge the gap across the topological transition. During the model development, we realized that the phase field we propose may add important features to the standard mesoscale models of bilayer membranes. In fact, we managed to include the Gaussian energy contribution and its reaction force in a smooth way into the model. From the mathematical perspective, this paved the way for using the well-developed machinery of variational analysis. The procedure is, at least conceptually, simple after leveraging on recent developments – the so-called string method – in the field of rare events. This allowed us to extract the free energy profile along the transition and to find that the Gaussian contribution and the *related reaction force* are crucial elements in establishing the free energy barrier.

Looking at the solution, we find that, approaching the critical state in the forward direction, many details are reminiscent of the intermediate states found in atomistic simulations, see the newly added Figs. 6, 7 and related comments. We speculate that elasticity may still play a significant role even in these extreme states.

- *“Incidentally Fig. 3 comforts the idea that the standard Canham-Helfrich description would also provide a rather accurate description of the topological transition considered here. It seems to this referee that the phase field method is indeed convenient as no cuts need to be introduced in the*

surfaces during fission or fusion, but does not fundamentally describe the actual physical process in a more realistic manner.”

From the fundamental, strictly rigorous point of view, we share the same conviction: our proposal can be considered as a way to allow the topological transition, leading to configurations that, far from the singular topology change, reproduce the Canham-Helfrich well-established description. Still, the phase field method combined with the proposed Gaussian energy functional apparently adds new features to the standard Canham-Helfrich model. In particular, it allows access to the force field associated with the topological constraint provided by the Gauss-Bonnet theorem. As shown in the new Figs. 6 and 7, this force contribution is crucial in the intermediate, stalk- and hemifusion-like, configurations.

- *“How “natural” or realistic is the phase-field method should also be assessed against the actual behavior of two vesicles pushed together: in most situations they repel and flatten at the contact point rather than fuse. Does then this method then describes membranes that always fuse and never repel each-other? If this is the case, what are the fundamental forces missing from the picture and how could they be re-introduced?”*

Clearly, the phase-field misses the atomistic details. Nevertheless, many aspects can be added and suitably tuned. For instance

- The bilayers of the two membranes can be made to interact through a suitable potential, as shown in Fig. 1 attached below. The interaction potential can be optimized to take into account adhesion or de-hydration effects, e.g., along the apposition phase of the two vesicles (see the new paragraph added in the Discussion section).
- The effective Gaussian modulus of the bilayer can be modulated – see also the answer to the comment referring to the fusogenic effect of monolayers’ negative spontaneous curvature and the paragraph added at the end of the Discussion section.
- The phase field can be used to separately describe the two mono-layers.

We like to stress that our vesicles do not spontaneously fuse unless the applied force is able to overcome the large energy barrier that we show in Fig. 2 of the paper. Note that the “shape” of the applied force plays a role, making the topology change easier if the external force reproduces the field we have calculated.

Figure 1: Preliminary (2D) simulation of two approaching vesicles showing flattening and adhesion.

- *“Although the forces acting on the $\phi=0$ surface present some interest for the description of the shape evolution, could the differential of forces acting on one side or the other of the dividing surface bring additional information helping to bridge the gap towards the molecular scales? It would be for instance interesting to know if such an indicator or other related ones can provide an upper cutoff for the forces needed to lyse the membrane and initiate a merging stalk.”*

Inspired by the Referee’s comment we have added more than one new column of text in the Discussion section and the already cited new figures 6 and 7 to discuss the issue. Fig. 6 shows the force spatial distribution along the steepest stretch of the MEP that is now significantly refined with respect to the previous version.

In addition, we have evaluated the force differential across the dividing surface, providing in the new fig. 7 its behavior along the string. Indeed, we find that the (external) force differential tends to lyse the membrane, as the Referee seems to suggest.

- *“It is well known that lipids with reverse spontaneous curvature have fusogenic properties, how well can this approach account for non-zero spontaneous curvature?”*

We thank the Referee for raising this interesting issue. It is known, see e.g. Ref. [12] in the bibliography of the paper, that the Gaussian modulus of the bilayer can be expressed in terms of properties of the two monolayers, $k_G = 2(k_G^{\text{ml}} - k^{\text{ml}}z_0m^{\text{ml}})$, where k_G^{ml} is the (negative) Gaussian modulus, k^{ml} the bending rigidity, m^{ml} the spontaneous curvature of the constituent (symmetric) monolayers, and z_0 the monolayer’s pivotal plane position. It follows that a negative spontaneous curvature of the monolayer tends to reduce the absolute value of the bilayer Gaussian modulus, $|k_G|$. The new inset of Fig. 2b shows that a reduction of $|k_G|$ leads to a reduced fusion barrier. Hence, the effect of the monolayers’ spontaneous curvature seems to be correctly reproduced by our model. We stress that two additional string computations were performed to check the effect of changing the Gaussian modulus.

Concerning the overall bilayer’s spontaneous curvature, Fig. 1 of the paper already shows the effect m may have on the transition. In addition, we have reported a case of spontaneous curvature-induced fission in the Supplementary Information.

- *“The inset in Fig. 2 (c) perturbs the understanding of the information in the figure, without much added value.”*

Fig. 2c has been modified, removing the inset and increasing the scale for better readability.

- *“Fig. 4 is not optimal. Its color scheme in the grey part is confined to a very small region (the interface) and thus it does not convey any easily visible information. Since the force field is obviously symmetric around the midplane, one can confine it to the lower part of the figure and use a vector representation in the upper part for the normal, this would be more informative than the color scheme.”*

We have modified Fig. 4. We decided to make it symmetric to better convey the geometry of the force distribution.

In conclusion, we take the chance to thank the Referee once again for the time they spent on our paper and for the extremely useful comments that allowed us to amend the text and better explain our point of view. In conclusion, we are convinced that the paper gained much after the referral and are now confident that it may be considered ready for publication in Communications Physics.

**Reply to the comments of
Referee 2
to the paper
Topological transitions in fluid lipid vesicles:
activation energy and force fields**

July 30, 2022

We thank the Referee for providing a detailed report on our paper and for the kind words of appreciation for our work, “The presented results in Figure 4 are very interesting, as it illustrates the spatial force acting on the membrane before and after the fusion process, which aligns with one’s intuition.”, “The outcomes of the study are rather interesting and the methodology is quite novel in the field ..”. However, the Referee raises a few general issues, “the study fails to address so many aspects of topological transitions”, that we answer in detail below before addressing the list of specific comments provided in the report.

- “ The report describes two genus-zero $g=0$ vesicles’ topological transformation to form one genus-zero $g=0$ vesicle. However, from the general title of the paper one would expect other examples such as $g=1$ to $g=0$ transformation and higher genus vesicles, as an example please see the preprint by the Jülicher group which reports topological morphogenesis of organoids: Ishihara, K., Mukherjee, A., Gromberg, E., Brugués, J., Tanaka, E. M., and Jülicher, F. (2021). Topological morphogenesis of neuroepithelial organoids. *bioRxiv*. ”

We thank the Referee for pointing our attention to this interesting paper, that we have read with interest and will, hopefully, get published soon. If we understand correctly, the Referee is complaining that we only considered one single example of topological transition, namely from two spheres to a prolate and vice versa. In order to show the generality of our approach, consistently with the proposed title of the paper, we now add to the Supplementary Information two additional cases, the transformation of an oblate vesicle into a toroid and of two nested spheres into a stomatocyte. In these cases, the topology change is triggered by an external force field, see Fig. 1 attached below.

Figure 1: Topological transition between an oblate vesicle and a torus, top, and from two nested spheres and a stomatocyte (bottom).

- “ Despite the novelty of the method used in the context of membrane biophysics as well as interesting results, I believe the results would not interest a wide range of audiences. The general

mathematical formalism seems rather complex compared to the simpler and widely used Helfrich model. One of the good features of the Helfrich model is the simplicity of the model not only for theoretical and computational adaptation but also for experimental studies. For instance, one can extract several essential parameters from the mean curvature of vesicles using a spherical cap approximation. Thus I doubt that the results of the study are simply reconcilable with the experimental investigations in the future. However, to strengthen the conclusion of the study I would encourage the authors to address and clarify a few points about the study and expand the study to a larger parameter space as mentioned in the following.”

We agree with the Referee that the Canham-Helfrich (CH) model is invaluable given the quality of its results combined with an unsurpassed simplicity that allows its use as a back-of-the-envelope calculator. We like to point out, however, that our contribution is intended to provide a richer, though undoubtedly more complex, description of the large-scale topological modifications of a membrane. Actually, strictly speaking, the CH model does not allow for topological transitions, given the singularity taking place when the membrane topology abruptly changes. This comment should not be intended to mean that the CH model cannot be cleverly used to obtain information on the topological rearrangement (see, e.g. Ref [72] in the paper bibliography). However, the importance of bridging the gap across the topology rearrangement via a smooth, technically rigorous approach should not be underestimated. Actually, by providing a thorough expression for the Gaussian energy contribution to the membrane elasticity, we have achieved to find the minimal energy path (MEP) that the membrane should follow to pass from one configuration to the other. The MEP provides the energy barrier that stabilizes the topology and gives access to the elastic reaction force of the membrane, showing the structure of the optimal external force field able to trigger the transition with minimal work expenditure. This result could simply be not achieved within the standard framework of the CH model, which cannot provide an indication of the force contribution arising from the topological constraint associated with the Gaussian energy. Frankly, we do not see any reason why the CH model should be reconcilable with experiments and ours should not, given that whenever the CH model works, ours provides exactly the same results, but generalizes to the transition. Of course, ours is not a fast solution and requires some skills in field theory and computational techniques. As a counterpart, it offers the unprecedented possibility to find the transition path and, we believe, will be of interest to a possibly large audience, including, but not limited to, the community working on membrane dynamics.

In the following, we eagerly try our best to clarify all the issues that were unclear in the original text.

- 1- *“One clear aspect is that a similar fusion process is not available for any arbitrary reduced volume simply due to geometrical constraints. It would be very useful if the authors can provide a supplementary figure or note discussing this geometrical aspect.”*

We are not completely sure we fully understood the question. As the Referee is certainly aware, the equilibrium configurations of a membrane are dictated by a certain number of parameters, in particular reduced volume and topological genus. Extensive literature discusses the phase space of equilibrium membrane configurations modeled by the CH Hamiltonian. All these configurations are perfectly reproducible by our approach. When considering topological transitions, one is asked to connect two (initial and final) configurations through a sequence of intermediate ones (in our jargon this sequence is called a string or a path). As the Referee will agree, it is of interest to identify the minimum energy path. In principle, given that compatible initial and final states exist, there is no limitation on the specific value of the reduced volume. Clearly, the constraint on membrane volume and area play a role in determining the MEP. In principle, the dependence on reduced volume can be parametrically explored. For the time being, we limit ourselves to demonstrating the generality of the approach by producing the MEP for a case where we completely relax the volume constraint, thus allowing the reduced volume to vary, see the new version of the Supplementary Information where we also better explain the value of the reduced volume addressed in the main text.

- 2- *“The Ginzburg-Landau phase-field employed in this study seems rather arbitrary and it is not clear how much the outcomes would change by changing the expression of the phase field. An elaboration regarding this issue is required.”*

The rationale is that the functional should converge to the Canham-Helfrich Hamiltonian in the limit of small membrane thickness. This aspect is thoroughly discussed in the Methods section and in the Supplementary Information where we have added a new paragraph discussing its relation

with possible alternatives and their drawbacks. Our new formulation allows a smooth description of the topological modification providing access to the membrane reaction force along the transition which would be otherwise inaccessible. Specifically, to the best of our knowledge, this is the first time that the energetic effect of the topological constraint is accurately taken into account in the evolution.

- 3- *“The claim made in the second column of page five “... fusion and fission processes cannot take place spontaneously and require further agents to happen, in addition to the elasticity and thermal fluctuation” does not seem accurate. Reference 25 is a contradictory example of this claim, where membrane-bound proteins only induced positive spontaneous curvature which can cause the division of a dumbbell shapes vesicle into two spherical vesicles. Such spontaneous curvature can be induced by exposing the vesicle to other forms of asymmetries.”*

The Referee is right! We have modified the text in order to be more precise about this point. We intended to convey the idea that in most cases of relevance for biology the membrane configuration is stable against topological transitions. In these cases, some external agent should act on the system, even via a change of spontaneous curvature.

- 4- *“The effect of spontaneous curvature is overlooked in the large part of the study; yet, spontaneous curvature is one of the most important material parameters that can induce fission and fusion. Also, the initial reduced volume of the vesicle is not defined for most cases.”*

In order to comply with the Referee’s explicit request, we now add a spontaneous curvature-induced transition of a prolate into two spheres in the Supplementary Information, as observed in the experiments described in the paper mentioned by the Referee, former Ref. [25], presently [35]. Please note the pearling-like instability illustrated in Fig. 1 of the original version of the paper was indeed a spontaneous curvature-induced transition. As the Referee is certainly aware, it is often the spontaneous curvature of the constituent monolayers that have fusogenic effects. At the level of the global membrane, this amounts to a reduction of the (absolute value of the) Gaussian modulus, as discussed in the present version of the Discussion section.

- 5- *“The gaussian modulus is assumed to be equal to the bending rigidity. This seems a plausible assumption. However, it would be worth showing how the energy landscape and force field of the fission and fusion changes if the gaussian modulus is larger or smaller than the bending rigidity.”*

We thank the Referee for the suggestion, we have performed new simulations for different values of the Gaussian modulus. We have added the corresponding energy profiles along the transition in the inset of the new version of Fig. 2b. The data show that the fission barrier is essentially unaffected, while the fusion barrier is strongly influenced. In particular a decrease of $|k_G|$ is found to reduce the fusion barrier, consistently with the fusogenic effect of negative spontaneous curvature of the monolayers forming the membrane, see also the last paragraph of the new version of the Discussion section. We find that also the force field along the fission branch of the MEP is unaffected by the change in the Gaussian modulus, consistently with the energy profile. On the other hand, we have indications – to be substantiated by further simulations – that along the fusion branch the forces are reduced by decreasing $|k_G|$.

- 6- *“The axial symmetry is assumed in the process. The effect of non-axially symmetric neck geometries is completely omitted. However, recent studies shoes that neck symmetry can break under different circumstances, for example, see. Vasan, R., Rudraraju, S., Akamatsu, M., Garikipati, K., and Rangamani, P. (2020). A mechanical model reveals that non-axisymmetric buckling lowers the energy barrier associated with membrane neck constriction. *Soft Matter*, 16(3), 784-797. Satarifard, V., Grafmüller, A., and Lipowsky, R. (2018). Nanodroplets at membranes create tight-lipped membrane necks via negative line tension. *ACS nano*, 12(12), 12424-12435.”*

We have found this comment especially interesting. For the time being, we have added a paragraph in the section devoted to the final comments where the related bibliography is now explicitly cited. In order to show the potentiality of our approach, we stress that the results of Fig. S5 in the Supplementary Information, although referring to an axisymmetric configuration, were obtained with a fully 3D code. Inspired by the Referee’s comment, we plan to study more in-depth non-axially symmetric neck geometries and related barriers in the near future.

- 7- *“The intermediate state of the membrane fusion is characterized by the formation of the hemifused state. From the presented results it was not clear if the phase-filed model can capture the existence of such a state.”*

In order to see these details, we have refined the string in the steepest stretch of the MEP. The intermediate states we find (see the new Fig. 6) are reminiscent of those observed in MD simulations and visualized in experiments measuring the electron density via X-ray diffraction. The spatial distribution of the force field at these intermediate states indicates that the external force should apply a force differential on the membrane able to induce its lysis, before the rearrangement that leads to the topological change. We address these issues in a new paragraph in the Discussion section together with the new Figs. 6 and 7.

- 8- *“One interesting morphology would be two nested vesicles where fusion would lead to stomatocyte morphology. In addition, a study of the topological transformation of spherical vesicle into toroidal vesicle would be very interesting.”*

Thank you for suggesting that. We have added new results concerning both configurations in the Supplementary Information, see also the figure reported above.

Here follow the answers to the final questions.

- 1- *“In the fission example (Figure 1), why do we only observe symmetric divisions?”*

Since the initial configuration is exactly symmetric and our code preserves the symmetries, the entire evolution remains symmetric. The symmetry may be broken in different ways, e.g. by assuming a closed end on one side and a reservoir on the other, like when pulling a membrane tether.

- 2- *“What is the effect of reduced volume on the fission-fusion force field?”*

In principle, our approach could provide an answer to this question. That would however require an entirely new simulation campaign, which at the moment is beyond the feasibility with the current time limits. It may probably be useful to stress here that each configuration along the string – a string consists of order 100 or more configurations – requires the relaxation of a full field which, though not unfeasible, is still significantly demanding from the computation point of view. Nevertheless, to partially address the Referee’s question, we have generated a new string removing the volume constraint. The corresponding results are now included in the new version of the Supplementary Information.

- 3- *“Did you study the hysteresis? How does the reversible pathway look like?”*

Before answering, allow us to reiterate the definition of Minimal Energy Path. According to the definition, reported in the paper, a Minimal Energy Path is a curve on the energy landscape $E[\phi]$ connecting the two stable states, and such that it is everywhere tangent to the gradient of the potential $\partial\phi/\partial\alpha \propto \delta E/\delta\phi$, except at critical points. In this sense, it is thermodynamically reversible and may be equally run in both directions. Hence, there is no hysteresis in the process. Typically, hysteresis takes place in this kind of system if along the evolution the free-energy barrier changes, making the reverse path different from the forward one, or if the path is influenced by inertial effects which is not the case for the example considered in the paper. Clearly, the barrier height may be so large that the system gets trapped for extremely long times in the metastable states. This is why we are stressing that external forces need to be applied to overcome the barrier, thermal fluctuations being unable to drive the transition in an acceptable time scale.

- 4- *“What would be the effect of non-zero temperature in the results of your study?”*

In principle, finite temperature effects may be included, see e.g. E, Ren, and Vanden-Eijnden, J. Phys. Chem. B, 2005. However, we do not expect significant finite temperature effects when the barrier is sufficiently large, like in the cases we are discussing.

- 5- *“How does the computational cost of your method compare to the computation of the Helfrich model by minimizing triangulated membrane or solving ODEs?”*

The question is hardly answered in general and would require a one-to-one comparison of the efficiency of the two approaches that we are planning to do in the near future. In principle, the operation count will depend on the number of degrees of freedom, N_{dof} . For a surface, $N_{dof} \propto 3S$, where S is the extension of the surface. For a phase field, which rapidly approaches ± 1 away from the membrane mid-surface, $N_{dof} \propto (l_{pf}/\epsilon) S$, where $l_{pf} = 6\epsilon$ is the interface thickness as defined in the paper. Without considering topological transitions, this is more or less the estimate

for an explicit algorithm, assuming that the same time step is employed. For our approach, we know that a substantial speed-up with respect to explicit algorithms is obtained using semi-implicit solvers, which render the terms with higher order derivatives unconditionally stable. Concerning triangulated Canham-Helfrich solvers, we expect, being no specialists in the technique, that the time step needs to be reduced, together with the mesh refinement, as more intense curvatures develop.

When it comes to evaluating the MEP across the topological transition and the membrane reaction force, that is the specific subject of the paper, we are convinced that our approach is especially able to deal with the problem.

In conclusion, we like to thank the Referee for their many insightful comments that allowed us to enlarge the scope of the paper and to make our contribution emerge in a much clearer way. We are now confident that they will be now get convinced to consider the new version of the paper ready for publication in Communications Physics.

**Reply to the comments of
Referee 3
to the paper
Topological transitions in fluid lipid vesicles:
activation energy and force fields**

July 30, 2022

We thank the Referee for the time spent on our manuscript and for the interesting comments and constructive criticism they rise.

From the report, we understand that we did not make ourselves fully understood. We confirm that the approach we propose strongly builds on the celebrated Canham-Helfrich Hamiltonian. Our original purpose was indeed to extend this classical approach to bridge the gap across topological transitions which are clearly hardly addressed in the context of sharp interface descriptions, without resorting to artificial surgery operations. After we succeeded in our intent (something not entirely trivial, we could say), we realized that the model we developed contains substantially new information strictly pertinent to the fusion/fission of lipid membrane, which we are convinced is worth being communicated to the community.

Of course, we do not claim that the model provides all the details of the process at the atomistic level. However, as this answer and the new version of the article are expected to clarify, many crucial aspects are well described and, moreover, the model lends itself to further improvements.

Following the Referee's comments, the paper has been significantly updated, including new figures based on entirely new simulations. For the Referees' convenience, the new version of the paper highlights in red the modified/new parts. We now address below each specific comment reporting in italics the original question by the Referee followed by our answer.

- *“An important aspect of the manuscript is a mathematical framework for calculating the minimum bending energy of a surface given a global surface area and volume limit. What is the purpose of focusing on the biomembranes and fusion process? Clearly, this creates a number of problems (see below). Compared to the Canham-Helfrich Hamiltonian (CHH), this approach also does not provide a correct energy landscape of the fusion process. For example, see S. Kawamoto et al., Free energy analysis along the stalk mechanism of membrane fusion. Soft Matter, 2014,10, 3048-3054 and C. S. Poojari et al, Free energies of membrane stalk formation from a lipidomics perspective Nature Communications 2, 6594 (2021). However, I see that the present study has “implications” for biomembrane shape, but provides no additional “reliable” information compared to CHH.”*

We thank the Referee for the very direct and frank comment on our proposal. Let us then try to summarize what is the novelty of the approach we propose and its relationship with the topological changes of lipid membranes. As the Referee points out, one of the features of our phase-field method is its ability to recover the Canham-Helfrich Hamiltonian. If this were the only feature of the approach we would agree that nothing substantially novel is proposed. There is however a crucial aspect that is totally new: the model is able to smoothly bridge the gap across topological transitions, which cannot be done based on the classic CHH approach. In doing so, we are able to include the crucial contribution of the Gaussian energy which, as we show in the paper, determines the shape and intensity of the reaction force field of the overall membrane and the height of the energy barrier that stabilizes the topology of the membrane. We were able to determine the minimal energy path also across the transition. Given the substantial height of the barrier, the transition goes through a sequence of non-equilibrium configurations and requires the action of an external force. In this respect, we like to stress that, although the approach allows finding the minimum

energy states, as the Referee acknowledges, the potential implications are much wider, e.g. in determining the optimal action needed of an external agent to induce the topology change.

The focus on lipid membranes and relative fusion/fission events arises from the need for a full-scale, all along description of the energetics guiding the process. Taking the Gaussian force into account leads to finding intermediate states in the merging region of the MEP (see the new Fig. 6) that, as the Referee will appreciate, are reminiscent of those observed in MD simulations (see the reference items kindly provided by the Referee, which we now include in the reference list together with others [23-25, 28, 29]) and visualized in experiments measuring the electron density via X-ray diffraction, [41]. In our case, none of the intermediates we find is a metastable state. Recent MD results, including those suggested by the Referee, show that the stalk and hemifusion states may not be (meta)stable in some cases. In particular, the stalk state may become unstable, increasing the initial distance between the disjointed vesicles, itself related to the hydration state of the bilayers, Refs. [25, 28]. We address all these issues in a new paragraph in the Discussion section together with the new Fig. 6 and 7.

The spatial distribution of the force field at these intermediate states indicates that the external force should apply a force differential on the membrane able to induce its lysis, before the rearrangement that leads to the topological change.

We fully agree that many important details at the molecular level are missing from the model. However, as we discuss in more complete form in the new version of the paper, there is substantial room for improvement. Moreover, after new simulations run on purpose to highlight the potential of our model and answer the Referees' criticism, we show that a change in the Gaussian modulus deeply influences the energy barrier in a subtle way: the barrier is unchanged in the fission direction, whereas it is significantly modified in the fusion direction, reduced, in particular, when $|k_G|$ decreases, see the inset of the new version of figure 2b. This is consistent with the fusogenic effect of lipids able to induce a negative spontaneous curvature of the constituent monolayers. In other words, the model contains much more physics than can be realized at first sight.

- *“Please elaborate on the rationale behind equations 5 and 7, particularly 7 since this is a term introduced in this work? Are these terms unique functional forms, or could there be other forms that give the same results? Is the field a measure of membrane density? Does it have any physical meaning? Would you have any suggestions as to how one can extend (fine tune) this term based on, for example, simulation data to capture the correct energy landscape? Also, does the method provide a unique numerical answer if there is no constraint on vesicle volume? Membranes are permeable to water, so not in all conditions, volume is constant.”*

We understand that the new functional expressing the Gaussian contribution to the membrane energy may look complicated. In fact, the overall problem has a long history in the field of variational calculus and differential geometry. After all, the rationale is that the functional converges to the standard form it takes in the Canham-Helfrich Hamiltonian in the limit of small membrane thickness. This aspect is thoroughly discussed in the Methods section and in the Supplementary Information where we have added a new paragraph discussing its relation with possible alternatives and their drawbacks. Our new formulation allows a smooth description of the topological modification (see also the next item) providing access to the membrane reaction force along the transition which would be otherwise inaccessible. Specifically, to the best of our knowledge, this is the first time that the energetic effect of the topological constraint is accurately taken into account in the evolution.

The direct physical interpretation of the phase field for a membrane is still under debate in the literature and no direct derivation from fundamental physics is known to us, e.g., by coarse-graining an atomistic model. Different interpretations can be proposed. One of them envisages the field as a normalized lipid density, Ref. [53]. On the other hand, a comparison with coarse-grained MD simulations seems to suggest that the field correlates with the water density.

Following the Referee's suggestion to discuss what would happen when removing the volume constraint, we have added new results to the Supplementary Information. We discuss the MEP for the transition between two disjointed vesicles and a single one sharing the area but leaving the volume free. Clearly, the minimal energy path identifies a well-defined evolution of the volume.

The model can be extended in several respects to include important features found in atomistic simulations, e.g.:

- i) The bilayers of the two membranes can be made to interact through a suitable potential. The interaction potential can be optimized to take into account adhesion or de-hydration effects, e.g., along the apposition phase of the two vesicles (see the new paragraph added in the Discussion section).
- ii) The effective Gaussian modulus of the bilayer can be modulated – see the paragraph added at the end of the Discussion section.
- iii) The phase field can be used to separately describe the two mono-layers.

Additionally, the full-scale evolution of a large membrane as described by the present model can be used as a base building block to realize multiscale simulations that can locally take into account the atomistic dynamics in the limited region where lipid rearrangement takes place.

- “*Topological transformation using Canham-Helfrich hamiltonian (CHH), can be captured through sudden moves, like the one presented in Gompper, G., and Kroll, D. M. (1998). Phys. Rev. Lett. 81, 2284–2287. The current work claims to have provided a smooth transition. However, in figure 2b, the number of the points in the transition region is only 4, and they are far (energetically) from each other.*”

Following the Referee’s indication, we now definitely show that the transition is smooth. For this purpose, we have refined the string (see the new Fig. 7) by putting one hundred points in the transition region. As already said above, this refinement allowed us to identify intermediate states during the fusion process that are reminiscent of the well-known pre-fusion, stalk, hemifusion, and fusion-pore states.

- “*On page 5, the authors write: $D=206$ nm is within the range of CHH validity, and well beyond the current limits of atomistic approaches. CHH becomes problematic when we deal with regions of high curvature; radii comparable to the membrane thickness. So, during membrane deformation, such regions could form even for extremely large vesicles (as it happens here too).*”

The Referee is indeed right! During membrane deformation, the membrane curvature may become large. At that stage, the CHH validity breaks down and it is exactly at this point that the diffuse nature of the field really enters into play taking into account the finite thickness of the bilayer, thus regularizing the solution. We stress that also during this phase the field behaves in a physically consistent way, see the new Figs. 6 and 7.

- “*Also, as to the second part of the statement: See for example below articles for current state of molecular dynamics simulations. They are not well below $D=206$ nm. Pezeshkian, et al, <https://doi.org/10.1101/2021.09.15.459697>; Pezeshkian, et al, Backmapping triangulated surfaces to coarse-grained membrane models; Nature Communications 11, 2296 (2020); J. V. Vermaas, Assembly and Analysis of Cell-Scale Membrane Envelopes J. Chem. Inf. Model. 2022, 3, 602–617.*”

To take into account the Referee’s comment, we have modified the Introduction which now contains the following paragraph: “The most commonly used techniques for in silico studies of topological transitions to date are coarse-grained molecular dynamics (MD) and dissipative particle dynamics (DPD) [19–29]. These computer simulations, which take into account the molecular details of lipid bilayers, allow monitoring in time morphological changes of small liposomes [30]. In many cases of interest, including topological transitions, the size of the vesicles is significantly larger or the characteristic time of the process is longer than accessible to purely atomistic methods. For example, in neurotransmission, the fusion of small synaptic vesicles can take hundreds of microseconds [31]. Recently, in order to reach larger vesicle sizes, hybrid, multiscale approaches have been proposed [32, 34]. Describing the complete topological rearrangement of these large-sized vesicles is the target of the present study.”

The references now include all the currently published papers the Referee is mentioning.

- “*Shape fluctuations (undulations) are an important feature of biomembranes (see for example L.Johannes, et al, Clustering on Membranes: Fluctuations and More Trends Cell Biol., 28 405-415 (2018)). What does equation 3 transform to for a flat membrane with small deviations? Does it capture the correct undulations spectrum?*”

We agree that, in general, membrane fluctuations may be important and we include the reference the Referee is mentioning in the new version of the paper.

The approach we propose can deal with thermal fluctuations by including in the evolution equation for the phase field a noise term designed to obey the fluctuation-dissipation balance. For the Referee’s convenience, we provide below a few still unpublished results that show that the equilibrium correlation – hence, the spectrum – is properly captured by our phase field formulation.

We like to stress however that the energy barriers we measure are sufficiently high to prevent thermally activated transitions.

Figure 1: Left: instantaneous configuration of the mid surface of a fluctuating planar membrane. Center: undulation spectrum. Right: reference (analytical) spectrum for the linearized Canham-Helfrich model.

Minor comments/suggestions

- *“CHH works for length scales much smaller than the one reported in the current manuscript: see Fiorin G, Direct derivation of free energies of membrane deformation and other solvent density variations from enhanced sampling molecular dynamics. J Comput Chem 2020, 41:449–459.”*

We thank the Referee for pointing our attention to that paper. We have amended the text accordingly.

- *“On page two, a sentence with “most of the current understanding is from the experiment”, contradicts the previous paragraph and is also not accurate.”*

We have modified the relevant text.

- *“ At the end of page 2, what does topological information mean?”*

Given a manifold, determining its topological genus is typically a nontrivial exercise, except by direct inspection in simple cases. By the Gauss-Bonnet theorem of differential geometry, the integral of the Gaussian curvature is a topological invariant which takes specific values depending on the genus of the manifold. In the mathematical paper “Retrieving topological information for phase-field models” the authors show that the integral of the Gaussian curvature can be retrieved from the phase-field description. We stress that our formulation differs from the one discussed there and, in our case, it is used not only as a topology descriptor but as a crucial dynamic ingredient of the membrane evolution.

- *“Numerous applications, what are these applications?”*

We now more explicitly describe the mentioned papers.

- *“ Membrane tension associated with area change (there are other types of membrane tension) is much greater than the bending energy. Is this not a better argument for a constant area algorithm?”*

Thank you for suggesting a better statement, which now replaces the original sentence.

We take the chance to thank the Referee once again for expressing in a crystal clear way their opinion, allowing us to better explain our point of view. In conclusion, we are convinced that the paper is now much more complete and clear, and are now confident that it may be considered ready for publication in Communications Physics.

REVIEWERS' COMMENTS:

Reviewer #1 (Remarks to the Author):

I would like to congratulate the authors for the careful and extensive revision of their work, convincingly addressing on the way a substantial amount of issues raised by this (and I believe others) referee(s).

I do now recommend publication of the manuscript

Reviewer #2 (Remarks to the Author):

Thanks to the authors for addressing all the concerns and questions and clarifying the ambiguities of the first draft. I recommend the publication of the paper in its current form.

Reviewer #3 (Remarks to the Author):

The authors have addressed my major concerns, and I do not need to review the article further. The current version of the article contains interesting results and the method has great potential. I recommend it for publication.

The following are two additional comments.

1) While the authors have thoroughly address my second comment in the rebuttal letter, they have not included all the responses in the main text. I strongly encourage them to do so. It may be included in the SI if the author believes that it will damage the flow of the main text.

In particular, I am referring to:

"Please elaborate on the rationale behind equations 5 and 7, particularly 7 since this is a term introduced in this work? Are these terms unique functional forms, or could there be other forms that give the same results? Is the field a measure of membrane density? Does it have any physical meaning?"

And the response from the rebuttal letter.

"

The direct physical interpretation of the phase field for a membrane is still under debate in the literature and no direct derivation from fundamental physics is known to us, e.g., by coarse-graining an atomistic model. Different interpretations can be proposed. One of them envisages the field as a normalized lipid density, Ref. [53]. On the other hand, a comparison with coarse-grained MD simulations seems to suggest that the field correlates with the water density."

"i) The bilayers of the two membranes can be made to interact through a suitable potential. The interaction potential can be optimized to take into account adhesion or de-hydration effects, e.g., along the apposition phase of the two vesicles (see the new paragraph added in the Discussion section).

ii) The effective Gaussian modulus of the bilayer can be modulated – see the paragraph added at the end of the Discussion section.

iii) The phase field can be used to separately describe the two mono-layers. Additionally, the full-scale evolution of a large membrane as described by the present model can be used as a base building block to realize multiscale simulations that can locally take into account the atomistic dynamics in the the limited region where lipid rearrangement takes place."

2) Why data points in Figure 7, are not included in Figure 2 as well?

Rebuttal Letter

Activation energy and force fields during topological transitions of fluid lipid vesicles.

September 14, 2022

We thank again all three Referees and willingly accept the advice of Referee 3 to include the discussion concerning the physical interpretation of the phase field and related issues that were originally included in our former rebuttal letter in the Supplementary Information. After a few trials, we decided not to add the symbols in the steepest part of the MEP of Fig. 2 to avoid blurring the image. Hence Fig.2 is left in its original form while Fig.7 describes the details of that region of the MEP.